# CD133-Dependent Activation of Phosphoinositide 3-Kinase /AKT/Mammalian Target of Rapamycin Signaling in Melanoma Progression and Drug Resistance

**DOI:** 10.3390/cells13030240

**Published:** 2024-01-26

**Authors:** Naji Kharouf, Thomas W. Flanagan, Abdulhadi A. Alamodi, Youssef Al Hmada, Sofie-Yasmin Hassan, Hosam Shalaby, Simeon Santourlidis, Sarah-Lilly Hassan, Youssef Haikel, Mossad Megahed, Robert T. Brodell, Mohamed Hassan

**Affiliations:** 1Institut National de la Santé et de la Recherche Médicale, University of Strasbourg, 67000 Strasbourg, France; dentistenajikharouf@gmail.com (N.K.); youssef.haikel@unistra.fr (Y.H.); 2Department of Operative Dentistry and Endodontics, Dental Faculty, University of Strasbourg, 67000 Strasbourg, France; 3Department of Pharmacology and Experimental Therapeutics, LSU Health Sciences Center, New Orleans, LA 70112, USA; tflan1@lsuhsc.edu; 4College of Health Sciences, Jackson State University, Jackson, MS 39213, USA; alamoudi.aa89@gmail.com; 5Department of Pathology, University of Mississippi Medical Center, Jackson, MS 39216, USA; yalhmada@umc.edu (Y.A.H.); rbrodell@umc.edu (R.T.B.); 6Department of Pharmacy, Faculty of Science, Heinrich-Heine University Duesseldorf, 40225 Dusseldorf, Germany; sofie00@gmx.de; 7Department of Urology, School of Medicine, Tulane University, New Orleans, LA 70112, USA; hshalaby@tulane.edu; 8Epigenetics Core Laboratory, Institute of Transplantation Diagnostics and Cell Therapeutics, Medical Faculty, Heinrich-Heine University Duesseldorf, 40225 Duesseldorf, Germany; simeon.santourlidis@med.uni-duesseldorf.de; 9Department of Chemistry, Faculty of Science, Heinrich-Heine University Duesseldorf, 40225 Dusseldorf, Germany; slh03122001@gmail.com; 10Pôle de Médecine et Chirurgie Bucco-Dentaire, Hôpital Civil, Hôpitaux Universitaire de Strasbourg, 67000 Strasbourg, France; 11Clinic of Dermatology, University Hospital of Aachen, 52074 Aachen, Germany; mmegahed@ukaachen.de; 12Research Laboratory of Surgery-Oncology, Department of Surgery, Tulane University School of Medicine, New Orleans, LA 70112, USA

**Keywords:** melanoma, CSCs, CD133, PI3K, AKT, mTOR

## Abstract

Melanoma frequently harbors genetic alterations in key molecules leading to the aberrant activation of PI3K and its downstream pathways. Although the role of PI3K/AKT/mTOR in melanoma progression and drug resistance is well documented, targeting the PI3K/AKT/mTOR pathway showed less efficiency in clinical trials than might have been expected, since the suppression of the PI3K/mTOR signaling pathway-induced feedback loops is mostly associated with the activation of compensatory pathways such as MAPK/MEK/ERK. Consequently, the development of intrinsic and acquired resistance can occur. As a solid tumor, melanoma is notorious for its heterogeneity. This can be expressed in the form of genetically divergent subpopulations including a small fraction of cancer stem-like cells (CSCs) and non-cancer stem cells (non-CSCs) that make the most of the tumor mass. Like other CSCs, melanoma stem-like cells (MSCs) are characterized by their unique cell surface proteins/stemness markers and aberrant signaling pathways. In addition to its function as a robust marker for stemness properties, CD133 is crucial for the maintenance of stemness properties and drug resistance. Herein, the role of CD133-dependent activation of PI3K/mTOR in the regulation of melanoma progression, drug resistance, and recurrence is reviewed.

## 1. Introduction

Human malignant melanoma is a highly aggressive skin cancer characterized by its heterogeneity, propensity to metastasize to distant organs, and the potential for developing resistance to conventional and even the newest targeted therapeutics as measured by progression-free and overall survival [1,2,3]. As a heterogeneous tumor, malignant melanoma exists in the form of genetically divergent subpopulations containing melanoma initiating cells/cancer stem-like cells (CSCs) as a small fraction, and non-cancer stem cells (non-CSCs) that form most of the tumor mass [4,5]. Like other CSCs, melanoma stem-like cells (MSCs) are characterized by their unique surface proteins and aberrant signaling pathways [4,5,6], which are either in a causal or consequential relationship to melanoma progression, treatment resistance, and recurrence [5,6,7,8]. CD133 (prominin-1) is one of the most important cancer stem cells (CSCs) that is widely expressed in the CSC subpopulation derived from a large variety of human malignancies, including melanoma [5,6,8]. Beyond its role as a reliable CSC marker for the identification of CSC populations [7,8], accumulating evidence has indicated that CD133 is responsible for CSC tumorigeneses and chemoresistance [4,8].

Aberrant activation of the phosphoinositide 3-kinase (PI3K) and mitogen-activated protein kinase (MAPK) pathways play a significant role in melanoma development, progression, and drug resistance. Melanoma is characterized by its heterogeneity as a consequence of genetic and non-genetic alterations that are frequently associated with the activation of the PI3K and MAPK pathways [9,10,11]. Although the dual inhibition of BRAF and MEK by their specific inhibitors has demonstrated significant treatment successes in advanced melanoma in patients with the BRAF mutation [12,13], most patients showed only short response durations along with the development of both intrinsic and acquired resistance [14]. The preclinical investigation of PI3K inhibitors as a monotherapy showed no significant advances in clinical trials [15,16]. The limited therapeutic efficiency of the inhibitors of PI3K pathway seems to result from the suppression of feedback loops mediated by PI3K/mTOR leading to the activation of compensatory pathways such as MAPK/MEK/ERK [17,18]. 

Melanoma treatment remains a major challenge in clinical oncology. Poor prognosis, disease progression, and drug resistance in melanoma are closely associated with the activation of PI3K [19,20]. Thus, targeting the PI3K/AKT/mTOR and MAPK/MEK/ERK pathways is a promising therapeutic approach that might improve treatment outcomes.

The main function of PI3K is to phosphorylate inositol-containing lipids (PtdIns) at the D3 OH group of the inositol ring. Consequently, the generation of PtdIns (3)P in the inner leaflet of membrane bilayers facilitates the recruitment of cytosolic proteins to initiate multi-functional signaling cascades [21].

Three different classes of PI3K have been identified [22,23]. Although the three PI3K classes, namely class I, class II, and class III, are similar in structure, they differ in their function. For example, class I of PI3Ks is directly involved in signaling downstream of plasma membrane-bound receptors, whereas the two other classes are involved in the regulation of vesicular trafficking [22].

The activation of PI3Ks is regulated by receptor–ligand ligation that in turn mediates the internalization of signaling receptors from the plasma membrane into early endosomes, where the receptors can be sorted into late endosomes/lysosomes for either degradation or recycling back to the plasma membrane [24,25]. Functionally, the PI3K class I is involved in the phosphorylation of PtdIns P2 to yield PtdIns P3 that is frequently altered in cancer, thereby serving as a potentially important therapeutic target [26,27,28]. 

Investigators have primarily focused on the heterodimer PI3K. It is composed of a catalytic subunit (P110) and a regulatory subunit (P85). The regulatory subunits contain non-receptor tyrosine kinase Src Homology 2 (SH2) and 3 (SH3) domains [29,30]. These SH2 and SH3 domains are characterized by their ability to interact with target proteins containing tyrosine kinase-binding sites [29,30].

It is notable that the activation of PI3K in many cancer types, including melanoma, is mostly associated with tumor development, progression, and drug resistance [31,32,33,34,35]. Melanoma frequently harbors genetic alterations in the key molecules of the PI3K pathway that can drive the aberrant activation of the PI3K pathway and its downstream pathways [27,36].

The conserved serine/threonine kinase mTOR, the mammalian target of rapamycin, is a downstream effector of the PI3K/AKT pathway and forms two distinct multiprotein complexes, mTORC1 and mTORC2 [9,10]. mTORC1 is sensitive to rapamycin and activates S6K1 and 4EBP1, both of which are involved in mRNA translation [37,38]. The activation of mTOR is mediated by diverse stimuli including growth factors, nutrients, energy, and stress signals, as well as signaling pathways, such as PI3K and MAPK [37,38].

Although the role of PI3K/AKT/mTOR in melanoma progression and drug resistance is well described, the inhibitors of the PI3K/AKT/mTOR pathway have demonstrated only limited success in clinical trials, particularly when applied as a monotherapy [35,36,37]. This shortcoming of PI3K/AKT/mTOR inhibitors as therapeutic targets in melanoma treatment is attributed to feedback loops derived from the inhibition of PI3K/mTOR leading to the activation/reactivation of aberrant signaling pathways like RAS/RAF/MEK/ERK [39,40]. 

This review provides insight into the role of CD133 signaling in the PI3K/mTOR pathway which can lead to melanoma progression, drug resistance, and recurrence, and the impact of the CD133/PI3K/AKT/mTOR pathway as a therapeutic target for melanoma treatment.

## 2. Melanoma Heterogeneity and Plasticity

Melanocytes are of neuroectodermal origin. They have the potential to migrate throughout the body and ultimately are localized in the skin, uveal tract, mucosa, inner ear, and rectum [41,42]. Based on the wide distribution of melanocytes, the development of melanoma can occur even outside these more common areas.

Despite access to early diagnostics and increasingly available access to primary care, both melanoma incidence and mortality rates are increasing worldwide. Current reports reveal that around 1.7% of all newly diagnosed skin cancers are melanomas, while patients dying from melanoma account for 75% of skin cancer deaths and nearly 0.7% of all cancer mortality [41,42,43,44,45,46,47,48,49,50,51,52,53,54,55,56,57,58,59,60,61,62,63,64,65]. Incidence rates differ significantly across the globe with the highest rates in Australia, New Zealand, Europe, and North America; the lowest incidence rate has been noted in Africa [46,47]. While the geographical dependency of melanoma development has been widely established, ethnicity, lifestyle (UV exposure), and genetic background are the most significant factors contributing to the development of melanoma [48,49]. 

Tumor heterogeneity is a significant challenge and is the main cause of drug resistance which ultimately leads to therapeutic failure [50,51,52,53,54]. It affects the therapeutic targets and shapes the tumor microenvironment influencing drug resistance [50]. Tumor heterogeneity is a tumor phenomenon that occurs spatially and temporally during tumor development with the aim of continuously triggering the reprogramming of the tumor microenvironment [50].

Tumor heterogeneity is attributed to the presence of subpopulations of cells that differ in their phenotype and biological behavior [56,57]. These subpopulations can exist within a tumor in the form of intra-tumoral or inter-tumoral heterogeneity. These two types of tumoral heterogeneity can occur in the same histopathological tumor subtype within a patient (intra-tumoral heterogeneity) or in the same tumor in different patients (inter-tumoral heterogeneity) [57,58].

The development of tumor heterogeneity in various cancer types including melanoma is mediated by genomic- (intrinsic factors) and non-genomic (extrinsic factors)-dependent mechanisms. Genomic tumor heterogeneity-dependent mechanisms are derived from significant alterations in the intrinsic factors including genome, transcriptome, epigenome, and proteome [60,61,62,63,64,65]. Consequently, melanomas are among the most common heterogeneous cancers. For example, melanomas are characterized by diverse genomic alterations in the form of functional mutations to NRAS, BRAF, KIT, CDK4, and MITF, loss of CDKN2A, PTEN, ARID2, and NF expression, and epigenetic changes to PTEN, CDKN2A, RAC1, and P53 genes [16,66,67]. In addition, the development of tumor heterogeneity in melanoma is the result of differential expression in the same tumor to various functional proteins, such as stem cell markers CD20, CD133, ABCB5, CD271, JARID1B, and ALDH1 [68,69,70].

Non-genomic (extrinsic factors) tumor heterogeneity-dependent mechanisms are the consequence of tumor-associated inflammation, and crosstalk between the tumor and its microenvironment [70,71]. Non-genomic/phenotypic heterogeneity demonstrates phenotypic variability between tumor subpopulations, which share the same genome, while trait differences among these tumor subpopulations are not related to the genetic differences between the identified tumor subpopulations [71,72,73]. In contrast to intrinsic factor-dependent mechanisms, the extrinsic factor-mediated tumor heterogeneity includes pH, hypoxia, and crosstalk between tumor cells and the components of the tumor microenvironment, particularly in association with stromal cells [50,71,72].

The contribution of inflammation to the development of tumor heterogeneity has been reported in several studies [74,75]. Inflammation is caused by a variety of pathogenic and environmental factors which lead to the induction of several mechanisms in the form of oxidative stress, upregulation of hypoxia inducible factor 1-alpha, and the production of pro-inflammatory cytokines [74,75,76,77]. 

The mechanisms involved in the development of phenotypic and functional heterogeneity of melanoma including genomic and non-genomic-dependent mechanisms are summarized in Figure 1.

In contrast to melanoma heterogeneity, the development of plasticity in melanoma cells is attributed to significant alterations in the expression profile of stromal and immune cells, and changes in the extracellular matrix of the melanoma microenvironment [75,78]. Cancer cell plasticity describes the ability of cancer cells to shift dynamically between differentiated and undifferentiated states to promote long-term tumor cell growth and transition into CSCs. 

Melanoma cells have the potential to switch their phenotype during tumor progression from a proliferative and differentiated phenotype to a more invasive and dedifferentiated phenotype [79,80]. Genetic alterations, influences from the tumor microenvironment, and epigenetic changes belong to the phenotypic plasticity and high heterogeneity that is known to be characteristic of melanoma [81]. Melanoma cells continuously undergo reversible alteration between a proliferative/differentiated and an invasive/dedifferentiated phenotype, an epithelial-to-mesenchymal transition (EMT)-like process [82,83]. Thus, the transition of melanoma cells into an invasive phenotype facilitates melanoma dissemination from a primary tumor to distant sites [84,85]. However, phenotype alteration is mediated mainly by a hypoxic tumor microenvironment and inflammatory signal-dependent mechanisms [85,86,87,88,89].

In addition to the reversible alteration between a proliferative/differentiated and an invasive/dedifferentiated phenotype [79,90,91], occurrence of melanoma cell plasticity is mediated by both cell-autonomous mechanisms and tumor microenvironment-dependent signals [90,92]. Accordingly, cancer cell/stromal cell-dependent mechanisms can impact the regulation of the phenotypic plasticity of melanoma cells [90,91,92,93,94,95]. For example, cancer-associated fibroblasts (CAFs), the major component in the tumor microenvironment, have been shown to play key roles in the regulation of melanoma cell plasticity via hepatocyte growth factor (HGF)-dependent mechanism [96,97] and insulin-like growth factor receptor signaling-dependent mechanisms [98].

Thus, understanding the biological complexity of cancer cell plasticity and its role in melanoma progression and relapse may lead to the development of new therapeutic approaches for the treatment of melanoma. Also, the functional analysis of specific cell markers and key molecules of aberrant signaling pathways is urgently needed, particularly those closely associated with the maintenance of stemness properties and the regulation of the cross talk between melanoma cells and their microenvironment. The mechanisms regulating melanoma plasticity are outlined in Figure 2.

## 3. Melanoma Stem Cells

The processes of melanoma development start in mature melanocytes [99,100]. Although the accumulated evidence indicates that early-stage precursors of melanocytes exist in the dermis [101,102], other reports have suggested that the earliest origins of cutaneous melanoma may have occurred in extrafollicular melanocyte stem cells [100,101,102]. This is probably because the melanoma stem cells (MSCs) are derived either from a transformed melanocyte, from a transformed melanocyte stem cell, or from a combination of both sources. Like CSCs, MSCs are characterized by their ability to self-renew and differentiate [103,104,105]. The generation of these subpopulations is mediated by genetic segregation and epigenetic alterations via the transcriptional regulation of genes associated with stemness properties and dysregulation of aberrant signaling pathways such as FOXM1 signaling [106,107,108]. Consequently, melanoma cells undergo intrinsically asymmetric cell divisions of stem cell lineage to produce two daughter cells, both significantly different in their genetic material and phenotype [109,110,111]. One of these daughter cells is characterized by its stemness properties and is referred to as CSC, whereas the other daughter cell (non-CSC) represents a larger portion of the overall tumor mass [4,5,9,110,111]. In addition to the deregulation of aberrant signaling pathways for tumor growth and survival, MSCs are notable for their expression of CD20, CD133, CD166, CD271, ABCB5, Nestin, and CD105 surface markers [4,5,8]. Thus, understanding the pathways controlling self-renewal, expansion, and differentiation of MSCs and how UV radiation alters and disrupts melanocyte lineage pathways will bring greater insight into the origin of melanomas. The development of CSCs from normal stem/progenitor cells (Figure 3A) and from cancer cells (Figure 3B) is outlined in detail in Figure 3.

Beyond their identification in various malignancies as part of the tumor mass, CSCs are characterized by their ability to confer self-renewal, differentiation, tumor initiation, metastasis, recurrence, and drug resistance [5,6,8,112]. Like CSCs of different tumor types, MSCs have been functionally characterized in vitro and in vivo [5,6,9,100,102]. In addition to their stemness properties, MSCs demonstrate the expression of stem cell markers including CD20, CD105, CD133, CD146, CD166, CD271, ABCB5, and Nestin [5]. Apart from their function as stem cell marker proteins, such as CD133, the proteins have also been discussed for their functional role in the regulation of MSCs maintenance and resistance [8].

## 4. CD133

The stem cell marker CD133 (prominin-1/AC133) has a molecular weight of 120 kDa and is encoded by the PROM1 gene [113,114]. It is a member of pentaspan transmembrane glycoproteins [114]. CD133 localizes specifically to cellular membranes with an extracellular N-terminal domain, 5-transmembrane domains separating two large glycosylated extracellular loops, two small intracellular loops, and an intracellular C-terminal domain [114,115]. The involvement of the CD133 protein in the maintenance of melanoma stemness properties and drug resistance is mediated through its C-terminal domain, which contains tyrosine binding sites located on tyrosine 828 (Tyr^828^) and tyrosine 852 (Tyr^852^) residues [9,116]. These two tyrosine residues are phosphorylation targets of the non-receptor tyrosine kinase (NRTK) Fyn [116]. The structure and the various domains of CD133 protein are illustrated in Figure 4.

Many studies have demonstrated that increased CD133 expression is associated with high tumorigenicity and metastatic potential for melanoma cells [117,118,119]. Also, CD133 protein has been implicated in the regulation of tumor resistance [120,121,122].

CD133-expressing CSCs have been shown to exhibit resistance to chemotherapy and radiation therapy in addition to being associated with poor prognosis in various cancers [122]. We and others have demonstrated that CD133^+^ cancer cells confer resistance to many chemotherapeutic agents such as caffeic acid phenethyl ester [5], Taxol [6], and fotemustine [8]. Accordingly, CD133-dependent mechanisms have been shown to be involved in the development of melanoma resistance to chemotherapy [8].

The contribution of CD133 to the regulation of CSC functions such as self-renewal, differentiation, and drug resistance are likely mediated by the NRTK, Fyn-dependent mechanism via the phosphorylation of the Tyr^828^ residue located on the cytoplasmic domain of CD133 [8,116]. Our laboratory has demonstrated that the phosphorylation of Tyr^828^ is essential for triggering the activation of PI3K and its downstream dependent signaling pathways in melanoma [8].

The PI3K/AKT pathway is one of the most important networks with the highest mutation frequency in human cancers [123,124,125]. Both PI3K/AKT/NF-κB and PI3K/AKT/mTOR are the two main mutated pathways involved in apoptosis and tumorigenesis facilitating the development of melanoma resistance to anti-cancer agents [126]. Dysregulation of major key molecules in these signaling pathways is associated with drug resistance and melanoma progression. Also, the elevated activation of PI3K pathway has been suggested as triggering melanoma progression through the activation of the PI3K/AKT/NF-κB axis [27,127,128].

In addition to the key role of CD133 in chemoresistance, we and others have demonstrated the cellular mechanisms by which the CD133 protein triggers activation of the PI3K pathway both in melanoma [8].

The PI3K/AKT/mTOR signal to downstream proteins leads to the development of tumor resistance [129,130]. Also, the PI3K/AKT/mTOR pathway has been shown to play a crucial role in a variety of biological and physiological processes including cell survival and growth, and transcription and translation, which are implicated in the development of drug resistance [18,33,131]. Abnormal activation of the PI3K/AKT/mTOR pathway in different tumor types including melanoma has been suggested as the key mechanism through which tumors evade drug toxicity [126,132,133]. Thus, CD133 mediates the activation of the PI3K/AKT pathway in melanoma [8,116]. The activation of mTORC1 through the PI3K-dependent activation of both AKT and PDK-1 has been reported in CD133 positive cells [8,116].

In addition to the frequent mutation to AKT family members, mutations to the PI3K/AKT/mTOR pathway are common in melanoma [33,34,35]. Mutation in PTEN has been shown to effectively restrain the PI3K/AKT/mTOR growth-promoting signaling cascade in primary and metastatic melanoma patients [131,132,133,134,135,136]. Also, the activation of PI3K/AKT results in the activation of mTORC1 that subsequently leads to the phosphorylation of the downstream molecules, p70S6K1, and eukaryotic initiation factor 4EBP1 to affect mRNA translation and protein synthesis [132,137]. The role of the Fyn-stimulated CD133 signal to PI3K is considered a key mechanism in the regulation of both PI3K/AKT and PI3K/NF-κB pathways. To that end, Fyn-stimulated CD133 signaling to PI3K is expected to trigger the activation of the mTOR and the associated biological consequences in melanoma. In addition, the references describing the functional role of Fyn-stimulated CD133 signal in melanoma have been mentioned (Table 1). The mechanisms are thought to be mediated by an Fyn-stimulated CD133 signal leading to the activation of PI3K and its downstream pathways, as shown in the proposed model (Figure 5).

## 5. Non-Receptor Tyrosine Kinases

In contrast to receptor tyrosine kinases (RTKs), non-receptor tyrosine kinases (NRTKs) do not possess either an extracellular ligand-binding domain or a transmembrane domain-spanning region [138,139,140]. Therefore, NRTKs are either localized in the cytoplasm [136], or anchored to the cell membrane through their amino terminal modification [141], or are underlying nuclear translocation [142]. NRTKs are characterized by their modular construction enzymes and individual domains that are connected by disordered regions in the form of loop connecting domains [140]. The catalytic domain of the NRTKs is of critical importance and is about 275 residues in length [143,144]. The structure of the catalytic domain is organized in two lobes. The small lobe functions as a binding site for ATP, whereas the large one functions as a binding site for the protein substrate [145]. The binding of both ATP and substrate to the corresponding domains catalyzes phosphate transfer in the cleft between the small and the large lobes [146]. In addition to their enzyme activity, NTRKs are characterized by their sequence preference around the target tyrosine [147]. The most Src-preferred sequence is Glu–Glu/Asp–Ile–Tyr–Gly/Glu–Glu–Phe, while the Abl-preferred sequence is Ile/Val–Tyr–Gly–Val–Leu/Val [148,149]. The difference in the preferred sequences around tyrosine in Src and Abl is evidence of their specific target substrates. NRTKs contain not only a tyrosine kinase domain, but also possess other domains that can mediate protein–protein, protein–lipid, and protein–DNA interactions [144,150]. Key protein–protein interaction domains in NRTKs are the Src homology 2 (SH2) and 3 (SH3) domains. SH2 is the longer domain of 100 residues and is characterized by the ability to bind, in a sequence-specific manner, phosphotyrosine (pTyr) but not unphosphorylated tyrosine (Tyr) residues [150,151,152,153]. In contrast to the SH2 domain, the SH3 domain is a small domain with 60 residues and is characterized by its ability to bind proline-containing sequences to form polyproline type II helixes [140,141,145]. While most NRTKs possess SH domains, there are also some that lack SH2 and SH3 domains [154,155]. These NRTKs possess some subfamily-specific domains that are essential for protein–protein interactions [156]. Among these subfamilies are the specific domains that target enzymes to the cytoplasmic part of the cytokine receptors [157,158], such as the JAK family, the integrin-binding domain, and the focal adhesion-binding domain (FAK family) [115,159]. The NRTK Abl also contains additional interaction domains including the F actin-binding domain and DNA-binding domain that contains a nuclear localization signal and exists both in the nucleus and cytoplasm [160,161,162]. Conversely, in addition to SH2 and SH3 domains, the NRTKs Btk/TEC subfamily possesses a pleckstrin homology (PH) domain [163,164]. PH domains are characterized by their ability to bind to phosphatidylinositol (PI) lipids [165,166].

PI lipids are key players in many trafficking and signaling pathways [167,168]. The phosphorylation of PI lipides is mediated primarily by a family of lipid kinases, the phosphatidylinositol-3-kinases (PI3Ks) [169,170,171].

PI3Ks are a family of enzymes that contain important cellular signaling regulators [18,172]. The activation of PI3K is mediated by either G-protein-coupled receptors or receptors with an intrinsic/associated protein tyrosine kinase activity in the form of an extracellular stimuli-dependent mechanism [173,174]. In addition to their activation by the direct interaction with the small GTPase Ras, the activation of PI3K by an NRTK-dependent mechanism has also been demonstrated in melanoma [175]. The structure of various NRTK families and their functional domains is illustrated in Figure 6.

## 6. Non-Receptor Tyrosine Kinase Fyn

Fyn is a tyrosine-specific phospho-transferase that is a member of the large Src family of NRTKs. While the formal crystal structure of the full-length Fyn protein has not been described, the mode of regulation of Fyn tyrosine kinase activity is like the Src family kinases [175,176]. Fyn is a 59 kDa protein comprised of 537 amino acids encoded by the Fyn gene that can be spliced to produce three isoforms [177]. The first identified isoform is isoform 1 (Fyn[B]); isoform 2 (Fyn [T]) is highly expressed in T-cells and differs from the isoform in the linker region between the SH2 and SH1 domains [178]. Isoform 3 has been detected in blood cells and differs from isoform 1 via the absence of sequence 233–287 [179,180]. Like members of the Src family, Fyn shares the conserved structure that consists of consecutive SH1, SH2, and SH3 domains (Figure 7). The SH1 domain is the catalytic tyrosine kinase, while the SH2 domain binds to tyrosine-phosphorylated substrates. Specifically, the SH2 domain of Fyn binds the phosphorylated tyrosine Y^528^ residue in its carboxyl terminal tail under basal conditions in vivo [116,181]. Repression of Fyn kinase activity is achieved via intra-molecular interactions between the SH3 domain and a polyproline type II linker helix that connects the SH2 and the SH1 domains. For the Fyn kinase, the tyrosine Y^528^ negative regulatory site is phosphorylated by C-terminal src Kinase (CSK), a cytoplasmic protein–tyrosine kinase [181]. FCSK homology kinase (CHK) is a second enzyme that catalyzes the phosphorylation of this inhibitory tyrosine Y^528^ [182]. CHK binds Src family members with a high affinity, independent of CHK catalytic activity, which may be sufficient to inhibit Src family kinase activity [183,184]. The dephosphorylation of the Y^528^ residue by protein tyrosine phosphatases rPTPα, SHP1/2, PTP1B, PTPε, and CD45 can release the SH2 domain and activate the enzyme [181,185,186]. In addition, the subfamily composed of Fyn, Src, and Lyn kinases contains dual acylation sites in the amino-terminal SH4 domain, which is thought to be partially responsible for lipid raft micro-domain association [181,186,187,188].

As is widely documented, the Src family kinase/focal adhesion kinase (FAK) complex is a signaling platform that is known to play a crucial role in the regulation of oncogenic growth factor receptor-dependent downstream pathways [189,190]. This observation is described more in the case of melanoma in Xiphophorus fish, in which the oncogenic EGF receptor orthologue Xiphophorus melanoma receptor kinase (Xmrk) effects the continuous activation of the Src family kinase Fyn that is strongly involved in promoting many tumorigenic events [190,191,192].

The Xiphophorus fish, which is derived from the crosses between X. maculatus (the southern platyfish) and X. hellerii (the green swordtail) species of the fish genus Xiphophorus [191,192], can spontaneously develop malignant melanoma via a proto-oncogene encoding a receptor tyrosine kinase-designated Xmrk-dependent mechanism [191]. The encoded Xmrk protein is structurally related to the human EGFR with an extracellular ligand-binding domain, a transmembrane domain, and intracellular.

The melanoma formation in Xiphoporus is initiated by overexpression of the EGFR-related receptor tyrosine kinase Xmrk. This receptor is activated in fish melanoma as well as in a melanoma-derived cell line (PSM) resulting in constitutive Xmrk-mediated mitogenic signaling. The elevated expression of Xmrk is the initial cell-type-specific event in melanoma formation in Xiphoporus [110]. Xmrk-mediated transformation potential is cell-type-specific signal transduction-dependent mechanism [193]. The NRTK Fyn has been identified as a substrate of Xmrk in Xiphoporus melanoma cells [181,194,195]. In addition, Xmrk has been reported to contain binding sites for growth factor receptor-bound protein 2 (GRB2), Src homology and collagen (Shc), Fyn, and PLCg [196]. Also, the binding of Grb2 to the activated Xmrk has been shown to trigger activation of the MAP kinase pathway [194]; meanwhile, the binding of Fyn to the Xmrk receptor is mediated through its SH2 domain [195].

FYN is highly expressed in many cancers and promotes cancer growth and metastasis through diverse biological functions such as cell growth, apoptosis, and motility migration, as well as the development of drug resistance in many tumors [12,151,177,178,179,180,181]. In addition, FYN is involved in the regulation of multiple cancer-related signaling pathways, including interactions with ERK, COX-2, STAT5, MET, and AKT [196]. FYN is therefore an attractive therapeutic target for various tumor types, and suppressing FYN can improve the prognosis and prolong the life of patients.

## 7. Receptor and Non-Receptor Tyrosine Kinase-Mediated Pathways to Melanoma Progression and Drug Resistance

In human melanomas, both PI3K/PTEN/AKT and RAS/RAF/MEK/ERK signaling pathways are the key routes whose constitutive activation results from genetic alteration [19,20]. Frequent mutations in RAF, RAS, and PTEN are largely associated with defects in cell death machinery, abnormal proliferation, angiogenesis, and invasion and thereby contribute in great part to melanoma progression and drug resistance [19,20,137,197].

The PI3K signaling pathway is one of the most frequently altered pathways in human cancer leading to oncogenic transformation, tumor initiation, and progression, in addition to the regulation of tumor apoptosis and autophagy [22,23,24]. Meanwhile, the PI3K and its lipid product phosphatidylinositol-3,4,5-trisphosphate (PIP3) are involved in the activation of multiple downstream signaling proteins [198]. Of note, the protein kinase AKT is the most studied downstream effector protein of PI3K [72,199,200].

The discovery of PI3K-dependent downstream effector signaling molecules such as the PDK1/mTORC2/SGK axis underscores the importance of PI3K in the regulation of a variety of cellular functions [200,201]. The PI3K/PDK1/mTOR/SGK pathway has been shown to compensate for the function of the PI3K/AKT pathway in the promotion of tumor survival, progression, migration, and drug resistance [20,82,202].

The target of rapamycin (TOR) was first identified in the budding yeast Saccharomyces cerevisiae [203]. The structurally and functionally conserved mammalian counterpart, mTOR, was identified based on the biochemical and inhibitory properties of rapamycin in mammalian cells [9,204,205,206].

mTOR includes two functionally distinct protein complexes, namely mTOR complex 1 (mTORC1) and mTOR complex 2 (mTORC2) [207]. mTORC1 is composed of mTOR, raptor, mLST8, and two negative regulators, PRAS40 and DEPTOR [207,208]. Meanwhile, mTORC2 is composed of the conserved mTOR, RICTOR (mAVO3), mSin1, and mLST8 (GβL), as well as less conserved proteins such as PRR5/Proctor, PRR5L, and DEPTOR [207,209,210].

The PI3K-Akt-mammalian target of rapamycin (mTOR) pathway is intracellular and is aberrantly upregulated in different tumor types, including melanoma [124]. The PI3K/AKT and PI3K/AKT-mTOR pathways are essential in the regulation of biochemical and biological processes both in normal and cancer cells [131,208,209,210,211,212]. Dysregulation of these pathways is frequently associated with genetic/epigenetic alterations and poor treatment outcomes in a variety of human cancers including melanoma [209,210].

The activation of PI3K through an NRTK/Fyn-induced CD133 signal has been reported both in melanoma [116] and glioma cells [211]. The phosphorylation of tyrosine kinase Tyr^828^ and Tyr^852^ residues by NRTK Fyn was found to trigger downstream pathways of PI3K, including PI3K/AKT/MDM2 and PI3K/AKT/MKP-1 [8]. Activation of the PI3K/AKT/MDM2 pathway results in the destabilization of the p53 protein, while the activation of the PI3K/AKT/MKP-1 pathway results in the inhibition of mitogen-activated protein kinases (MAPKs) JNK and p38, and the activation of both PI3K/AKT/MDM2 and PI3K/AKT/MKP-1 pathways leads to the inhibition of fotemustine-induced apoptosis [8]. Although the characterization of Fyn-stimulated CD133/PI3K/mTOR is not described in human melanoma, the mechanism of Fyn-stimulated CD133 signal to the PI3/AKT axis seems to be a general mechanism in solid tumors [116,211]. However, the activation of PI3/AKT is expected to mediate the Fyn-stimulated CD133 signal to mTOR in the form of generating an Fyn/CD133/AKT/mTOR axis. Thus, targeting the Fyn/CD133/AKT/mTOR axis likely possesses a therapeutic impact for melanoma treatment. Figure 8 demonstrates a proposed model for both RTK- and NRTK-mediated signaling pathways, PI3K/AKT, PI3K/AKT/mTOR, and RAS/RAF/MEK/ERK, in different tumor types including melanoma.

The activation of mTORC1 is mediated by the inactivation of both TSC1 and TSC2 following their phosphorylation via the PI3K/AKT pathway [205]. In addition to its functional role in the regulation of cell growth, proliferation, and survival in response to sensing mitogen, energy, and nutrient signals, the mTORC1 is involved in the regulation of many functional proteins including the regulation of eukaryotic elongation factor 2 (eEF2) kinase [212,213], CLIP-170 (cytoplasmic linker protein-170) [214], ornithine decarboxylase (ODC) [215], hypoxia-inducible factor 1α (HIF-1α) [216], protein phosphatase 2A (PP2A) [217], lipin [218], PKCδ and PKCε [219], protein phosphatase 2A (PP2A) [2], p21Cip1and p27Kip1cyclin-dependent kinase inhibitors [220,221], and retinoblastoma protein (Rb) [222].

The second complex of mTOR, namely mTORC2, has been demonstrated as playing an important role in the promotion of cancer cell survival, proliferation, growth, and motility based on its ability to enhance the phosphorylation of Akt^Ser473^, the key regulator of the insulin/PI3K pathway [9,223]. To that end, the permanent activation of AKT is associated with feedback reactivation of mTORC2 [224]. Thus, targeting mTORC1 and/or mTORC2 may have a therapeutic impact on the treatment of different tumor types, including melanoma. The mechanisms of RTK-mediated signaling to the PI3K/AKT/mTOR pathway and its biological consequences in melanoma cells are outlined in Figure 9.

## 8. Melanoma Progression and Drug Resistance Are Attributed to Dysregulation of PI3K/AKT/mTOR Pathways

Dysregulation of the PI3K/AKT/mTOR pathways contributes to the pathogenesis of melanoma [210,225]. The initiation of primary melanoma occurs as horizontal lesions that appear in the form of a thin plaque on the epidermis [226], and therefore is known as the radial growth phase (RGP) that can proceed to form the vertical growth phase (VGP) characterized by its invasiveness and propensity to metastasize to distant organs. The transition of RGP into VGP has been reported to be mostly associated with AKT activation, since the occurrence of melanoma metastasis is frequently associated, to a large degree, with AKT/mTOR activities [227]. Thus, the enhancement of the aggressiveness and metastasis of melanoma has been suggested to occur through AKT-dependent mechanisms [228,229]. In this process, AKT acts as a molecular switch that is linked with the elevated activation of mTOR and S6K1, enhancement of angiogenesis, and the accumulation of reactive oxygen species (ROS) that can enhance further aggressiveness and metastasis in melanoma [209].

As is widely documented, the development of melanoma is attributed, in great part, to intensive skin exposure to both UVA and UVB rays that contribute to the dysregulation of the PI3K/AKT/mTOR pathway [230,231]. Thus, abnormal cellular alterations in response to genetic and non-genetic modifications of genes or proteins of cutaneous intracellular networks are expected to destroy skin tissue homeostasis, and ultimately lead to the development of different types of skin cancer [82,232].

Accumulated evidence indicates that activation of the mTOR pathway is strongly associated with the pathogenesis of melanoma [40,233]. Meanwhile, constitutive activation of mTOR has been reported as inhibiting autophagic cell death and dysregulation of the normal cell cycle [234,235]. Given that the molecular basis and targets of most melanomas have been described in detail, the development of anticancer agents with the ability to maintain skin tissue homeostasis and integrity is a promising therapeutic strategy. Overall, targeting PI3K, AKT, and mTOR proteins by a specific inhibitor may offer a therapeutic option for the treatment of different tumor types including melanoma.

## 9. PI3K/AKT/mTOR Pathway as a Therapeutic Target in Melanoma Treatment

Successful prevention of tumor development and progression requires the long-term administration of anticancer agents that have to be well tolerated by patients. Previous and current analyses of signaling pathways in different tumor types have revealed that the tumor driving-signaling pathways of melanoma contain key molecules that can serve as therapeutic targets for melanoma treatment [236,237,238].

In addition to limited clinical benefit as a monotherapy in melanoma treatment, treatment with PI3K-targeted inhibitors is associated with the development of intrinsic and acquired resistance [239,240]. The development of these resistance mechanisms has been shown to be the consequences of PI3K inhibitor-induced insulin release, leading to hyperphosphorylation of insulin growth factor receptor(s) (IGF1R) [241,242]. As consequence, the phosphorylated IGF1R becomes able to mediate the reactivation of the PI3K signaling axis in tumors and rescues AKT and S6 phosphorylation via insulin receptor substrate (IRS) adaptor molecule-dependent mechanisms [241,243,244]. Insulin-mediated feedback loops can evade the biological consequences of PI3Kα, mTORC1, and AKT inhibition [17,245,246].

Both PI3K/AKT/mTOR and RAF/MEK/ERK signaling cascades are derived from numerous feedback loops and are interconnected at multiple points of crosstalk. Inhibition of one of these pathways can result in the activation of the other signaling cascade [247]. Thus, dual targeting of both pathways may improve treatment efficacy, leading to better clinical outcomes. Numerous in vitro and in vivo preclinical studies have revealed that dual targeting of signaling pathways, such as the PI3K/AKT/mTOR and RAF/MEK/ERK pathways, is a clinically relevant treatment option [248]. For example, pyridinyl imidazole compounds have been reported to simultaneously target both the BRAF oncogene and mTORC1 signaling in human melanoma cells [249].

Although the immune checkpoint inhibitors have been utilized as an alternative therapeutic option for melanoma treatment [250,251,252], tumor microenvironment-induced immunosuppressive effects have been shown to impair the therapeutic efficacy of immune checkpoint inhibitors [253]. In contrast to inhibitors of the immune checkpoint, PI3K/AKT/mTOR and MAPK/MEK/ERK inhibitors have been shown to suppress tumor growth and block tumor microenvironment-mediated tumor growth and metastasis. For example, the inhibition of PI3K/AKT/mTOR was found to increase immunogenicity and thereby enhance tumor sensitivity to immunotherapy [254]. Thus, the combination of MAPK/MEK/ERK- and PI3K/AKT/mTOR-targeted therapies may overcome resistance to immunotherapy [255,256]. Combinatorial or sequential treatments are expected to benefit melanoma patients, particularly those with mutations in both PI3K and MAPK pathways [255,256].

Many of the multiple targeted inhibitors for Fyn/CD133, PI3K, AKT, or mTOR have been evaluated and tested in in vitro melanoma model and in phase I and II clinical trials in melanoma. These include the inhibition of the Fyn/Stat pathway by chalcone derivative in melanoma [257], the inhibitor of Fyn/CD133 (Saracatinib) that has been evaluated via a Phase II study in metastatic melanoma [258], the inhibitors of PI3K (BAY-80-6946) [259] and of buparlisib [260], the inhibitors of AKT including perifosine in patients with metastatic melanoma [261], and MK-2206 in BRAF wild-type melanoma [262]. Meanwhile, the dual inhibitor of PI3K/mTOR (NVP-BEZ235) has been studied in melanoma [263]. Also, the mTOR analogs including everolimus [264] and temsirolimus [265] have been tested on their inhibitory effect on mTORC1 in melanoma; conversely, mTOR kinase inhibitors were found to target both mTORC1 and mTORC2 [266,267]. In addition, the most common references describing the reliability of the Fyn/CD133/PI3K/mTOR pathway as a therapeutic target in melanoma treatment have been provided in Table 2 and the possible therapeutic targets of the Fyn/CD133/PI3K/Akt/mTOR pathway in melanoma are outlined in Figure 10.

## 10. Conclusions

Malignant melanoma is a deadly disease with a poor prognosis. Obtaining complete tumor remission is difficult because of the presence of a heterogeneous subpopulation of CSCs. However, the identification of CSCs in melanoma and other cancers has led to promising advances that may soon impact the management of these cancers. CSCs are also responsible for therapeutic resistance that leads to tumor relapse. Specific signaling mechanisms are required for the maintenance of CSCs in tumors that can maintain their microenvironment. CSCs are becoming priority targets for the development of novel antitumor therapy. The tumor milieu is a critical regulator of melanoma-specific CSC-driven angiogenesis and metastasis. Signaling effectors from ECM or stromal cells can act as EMT or MET inducers or may regulate dormancy at metastatic sites in CSCs. CSC-dependent melanoma progression is mediated by MAPK/ERK and PI3K/Akt/mTOR pathways. Given the unique biology of CSCs, there is great need to develop novel and promising approaches for CSC-targeted cancer therapy. Combinatorial and/or sequential inhibition of CD133 signaling to PI3K/AKT/mTOR and PI3K/RAS/RAF/MEK/ERK pathways, after first-line immunotherapy, may extend the anti-tumor response for melanoma patients. This is especially true for those harboring genetic alterations in key molecules of both the PI3K/AKT/mTOR and MAPK/MEK/ERK pathways. The development of clinically relevant pharmacological inhibitors to block the function of both PI3K/AKT/mTOR and MAPK/MEK/ERK could provide new avenues for well-designed studies that assess the tolerability and efficacy of a new therapeutic approach for the treatment of melanoma.

## Figures and Tables

**Figure 1 cells-13-00240-f001:**
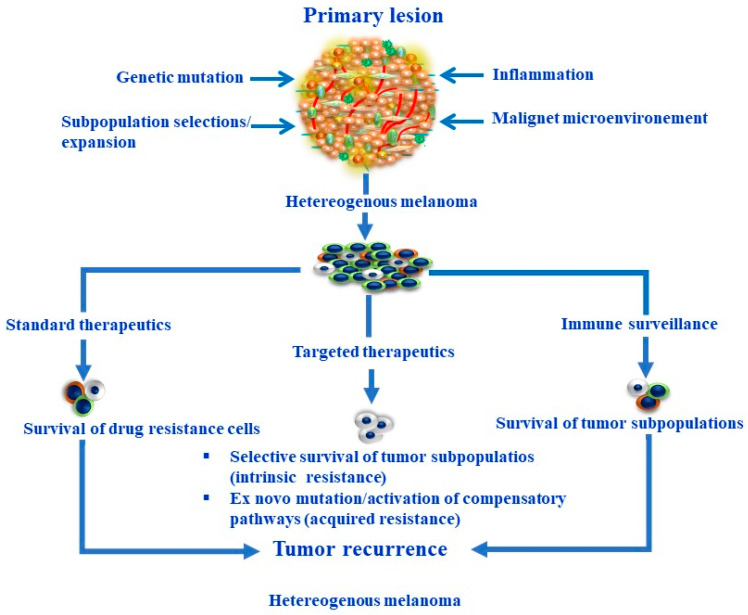
Overview of the genomic and non-genomic-dependent mechanisms regulating the development of melanoma heterogeneity and the biological consequences.

**Figure 2 cells-13-00240-f002:**
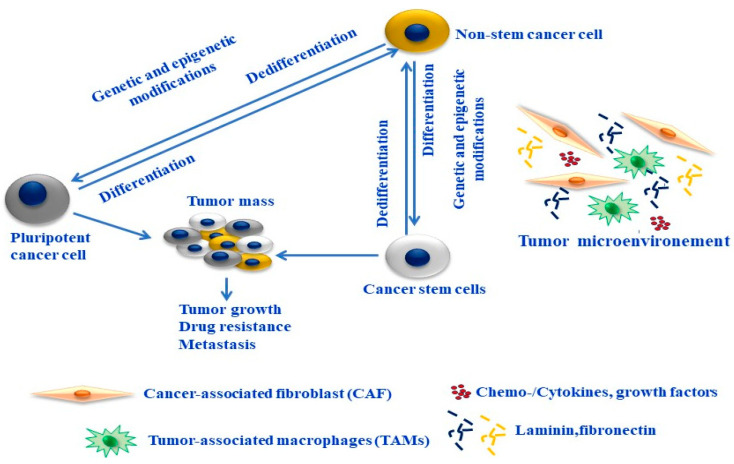
Cellular plasticity in melanoma. Melanoma cells can switch between a differentiated/proliferative/cancer stem cells (CSCs) and a dedifferentiated/invasive phenotype/non-stem cancer cells via mechanism-mediated. Melanoma plasticity is mediated by genetic and epigenetic alterations of melanoma cells and by tumor microenvironmental secretory products (e.g., growth factors and cytokines), activation of cancer-associated fibroblasts (CAFs), or tumor-associated macrophages (TAMs). Cancer cells can be reprogrammed towards pluripotency. Phenotype switch of melanoma increases their plasticity and is responsible for tumor growth, invasion, and drug resistance.

**Figure 3 cells-13-00240-f003:**
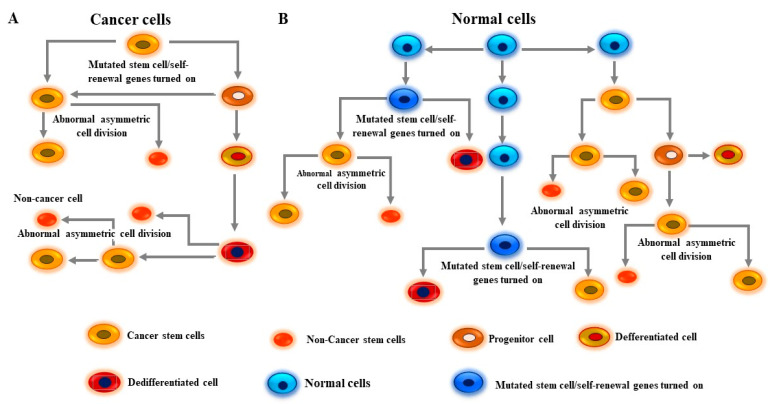
Model for the development of cancer stem cells (CSCs) from both tumor and normal cells. (**A**) Development of CSCs from cancer cells is mediated by the activation of aberrant signaling pathways via driver mutation-dependent mechanisms and transcriptional activation of self-renewal genes. Cancer cells become programmed to divide into cancer progenitor cells and CSCs. Cancer progenitor cells are genetically programmed to divide into one differentiated cell and one CSC. Once the dedifferentiation process of the differentiated cell has been completed, it can be transformed into a CSC that ultimately undergoes abnormal asymmetric cell division to produce genetically divergent subpopulations including CSCs and non-CSCs. (**B**) The development of CSCs from the activation of self-renewal genes normal stem/progenitor cells via multiple genetic mutations and dedifferentiation-dependent mechanisms. Like cancer cell-derived CSCs, CSCs derived from normal stem/progenitor cells undergo abnormal asymmetric cell division to produce two genetically divergent subpopulations including CSC and non-CSC.

**Figure 4 cells-13-00240-f004:**
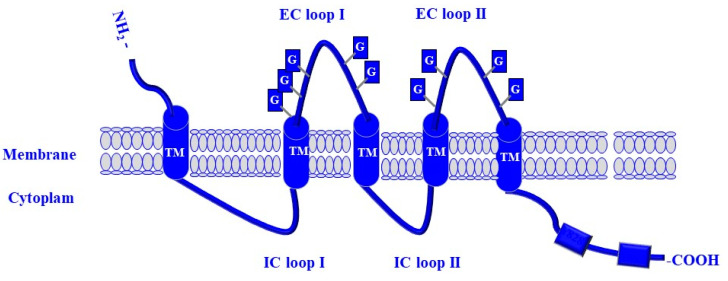
Structure of CD133 protein. CD133 is a 97 kDa transmembrane glycoprotein that contains an extracellular N-terminal domain, five transmembrane (TM) segments, two small intracellular loops (IC I and IC II), two extracellular loops (EC I and EC II), and an intracellular C-terminal domain. The two extracellular loops contain nine N-linked glycan (G) residues. The EC loop I contains 5 N-glycosylation sites, while EC II contains 4 glycosylation sites. The intracellular C-terminal domain contains two Tyrosine (Tyr^828^) and (Tyr^852^) residues.

**Figure 5 cells-13-00240-f005:**
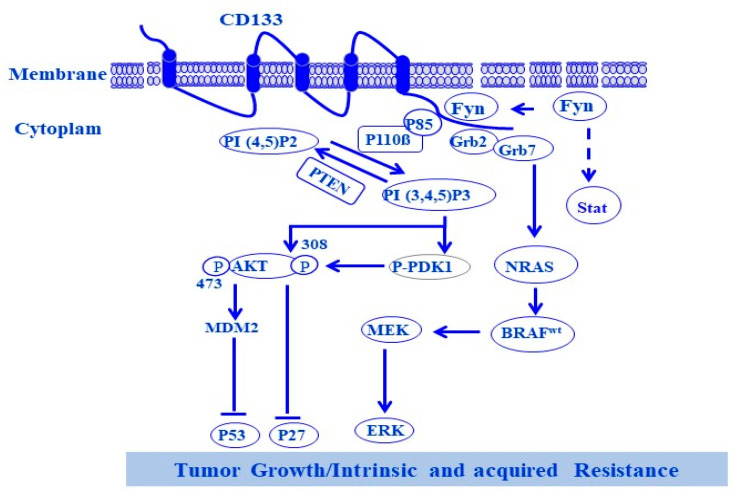
Function of CD133 protein. Phosphorylation of Tyr^828^ and Tyr^852^ residues of CD133 by non-receptor tyrosine kinase (NRTK) Fyn triggers the activation of PI3K- and NRAS-dependent pathways in melanoma. Upon the phosphorylation of both Tyr^828^ and Tyr^852^ residues, located on the cytoplasmic N-terminal domain of CD133, the phosphorylated Tyr^828^ (pTyr^828^) and Tyr^852^ (pTyr^852^) recruit the regulatory subunit of PI3K, p85, and the adaptor protein Grb2 to mediate the activation of PI3K/PDK-1/AKT and PI3K/AKT/MDM, and NRAS/RAFMEK pathways, respectively. The activated PI3K/PDK-1/AKT pathway induces the inhibition of p27 and the activated PI3K/AKT/MDM2 pathway induces the ubiquitination of p53. Meanwhile, the activated RAS/RAFMEK pathway induces the activation of ERK. Consequently, the activation of Fyn-stimulated signaling to downstream pathways of both PI3K and Grb2 results in melanoma growth and resistance.

**Figure 6 cells-13-00240-f006:**
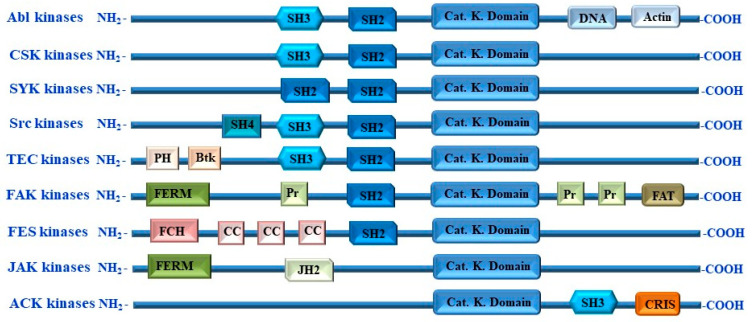
Structure of non-receptor tyrosine kinase families and their functional domains. Non-receptor tyrosine kinases (NRTKs) such as ABl, CSK, SYK, Src, TEC, FAK, FES, JAK, and ACK kinases are made up of an N-terminal region and a large C-terminal region. The N-terminus contains the common kinase domain that spans ~300 residues. The N-terminal region of NRTKs contains a number of extra SRC homology 4 (SH4), 3 (SH3), and 2 (SH2), catalytic kinase (SH1), Pleckstrin homology (PH), four-point-one, ezrin, radixin, moesin (FERM), Janus homology (JH2) domain 2, and FES/FER/Cdc-42-interacting protein homology (FCH) domains, with Btk-type zinc finger (Btk), coiled coil (CC) motives, proline-rich region (pr), Janus homology domain 2 (JH2). The C-terminal region of NRTKs contains SH3, CRIB, and Cdc42/Rac-interactive (CRIB) domains as well as DNA-binding (DNA), actin-binding (Actin), and focal adhesion targeting (FAT) domains.

**Figure 7 cells-13-00240-f007:**
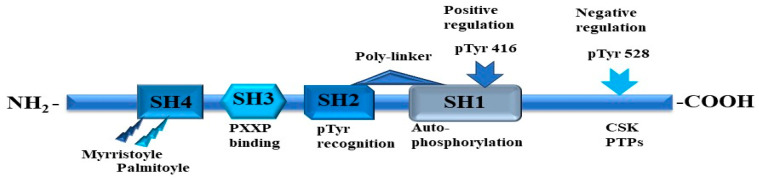
Fyn kinase structure and regulation. Fyn kinase consists of SH1, SH2, SH3, and SH4 domains. The SH2 domain binds the phosphorylated Tyr^528^ (pTyr^528^) in the C-terminus to keep Fyn in an inactive conformation. Tyr^528^ is dephosphorylated by phosphatases (PTPs) to keep the structure open allowing for phosphorylation of Tyr^416^ in the catalytic SH1 domain.

**Figure 8 cells-13-00240-f008:**
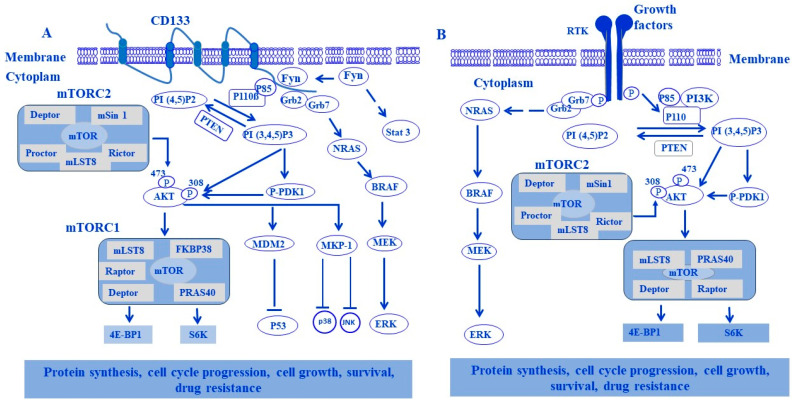
Proposed model for models of both non-receptor tyrosine kinase- (NRTKs) and receptor tyrosine kinase-mediated signaling pathways in tumor progression and drug resistance. (**A**) NRTK-mediated signaling pathways to tumor progression and drug resistance. NRTK Fyn stimulates Stat3, CD133 signaling to PI3K/AKT/mTOR, PI3K/PDK-1, PI3K/AKT/MDM2, and Grb2/Grb7/RAS/RAF/MEK/ERK pathways in melanoma by the phosphorylation of tyrosine kinase Tyr^828^ and Tyr^852^ residues located on the cytoplasmic C-terminal domain of the CD133 protein to recruit both p85 and Grb2 proteins, respectively. P85 serves to mediate Fyn-stimulated CD133 signal to PI3K to enhance the activation of several pathways including PI3K/PDK-1/AKT/mTOR/S6K and/or PI3K/AKT/mTOR/S6K; PI3K/PDK-1/AKT/4E-BP1 and/or PI3K/AKT/4E-BP1; PI3K/PDK-1/AKT/MDM2 and/or PI3K/AKT/MDM2; PI3K/PDK-1/AKT/MKP-1 and/or PI3K/1/AKT/MKP-1. Meanwhile, Grb2serves to mediate Fyn-stimulated CD133 signal to enhance the activation of NRAS/BRAF/MEK/ERK pathway. Finally, the biological consequences of Fyn-stimulated signal to CD133 in melanoma include the enhancement of protein synthesis, cell cycle progression, cell growth, survival, and drug resistance. (**B**) RTK-mediated signaling pathways to tumor progression and drug resistance. By binding RTKs to their corresponding receptors, tyrosine kinases (TKs) bind ligands to initiate the signaling pathway via intermediate molecule, insulin receptor substrate (IRS). Then, the activated PI3K becomes able to phosphorylate the phosphatidylinositol 4,5-bisphosphate (PIP2) to phosphatidylinositol 3,4,5-trisphosphate (PIP3), a process that is reversed by PTEN. Meanwhile, at the cell membrane, proteins with pleckstrin homology domains are then phosphorylated via PIP3 (phosphoinositide-dependent protein kinase-1 (PDK1) and AKT). Also, PDK1 can phosphorylate critical residues on AKT. The tumor suppressor complex of TSC1/TSC2 normally inhibits mTOR activation via the brain-enriched Ras homologue (Rheb). Activated AKT prevents this inhibition and leads to activation of the mTOR/Raptor complex 1 (mTORC1). This complex can be inhibited by rapamycin and its analogues. Consequently, mTORC1 leads to the activation of downstream proteins involved in the initiation of protein synthesis, leading to cell growth. Also, activation of RTKs able to initiate MAPK pathway signaling, leading to tumor progression, proliferation, and drug resistance. Activation of the MAPK pathway may also enhance PI3K signaling. MEK, MAPK/ERK kinase.

**Figure 9 cells-13-00240-f009:**
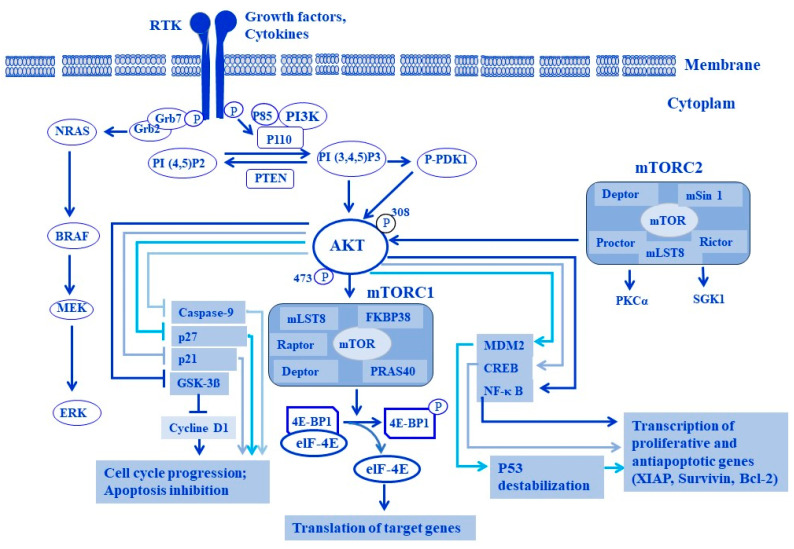
Downstream effectors of RTK-mediated pathways and their biological consequences. Upon receptor tyrosine kinase (RTK) stimulation by their corresponding ligands, both PI3K/Akt/mTOR and Grb2/RAS/RAF/MEK/ERK signaling pathways can be activated in melanoma. The activation of PI3K/AKT and NRAS/RAF/MEK signaling pathways by RTKs can promote several cellular functions via the activation of MDM2, NF-κB, and CREB to enhance p53 degradation as well as promote the transcription of proliferative and anti-apoptotic genes. PI3K/AKT-induced activation of mTOR phosphorylates 4E-BP1, the inhibitor of elF-4E, to allow the release of the active form of elf-4E to initiate the translation processes of angiogenic or cell cycle gene targets. PI3K/AKT can inhibit the phosphorylation of downstream effectors, namely caspase-9, GSK-3ß, p27, and p21 to induce cell cycle progression and inhibition of apoptosis.

**Figure 10 cells-13-00240-f010:**
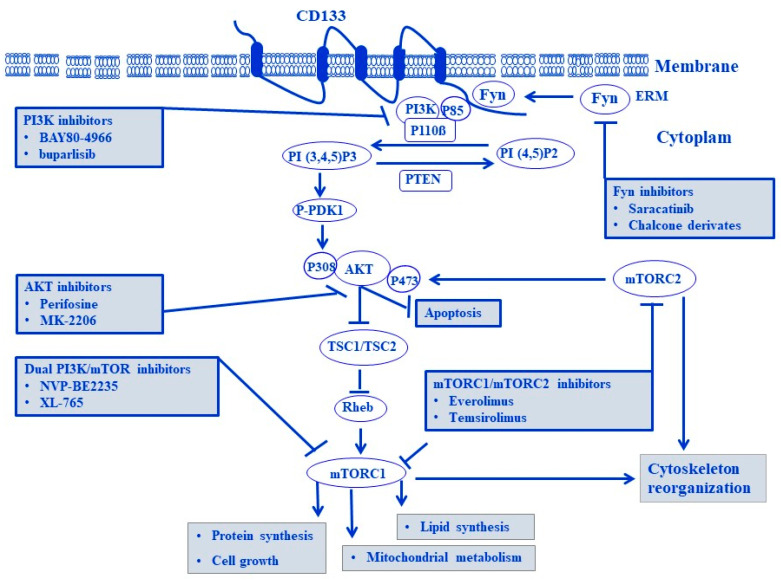
Selected agents targeting the Fyn/CD133/PI3K/Akt/mTOR pathway. Fyn, PI3KAKT, mTORC1, and mTORC1/mTORC2 inhibitors.

**Table 1 cells-13-00240-t001:** Studies focused on CD133-dependent P13K/AKT pathway signaling and its link to mTOR activation in melanoma progression.

Description of the Study	References
Melanoma progression and treatment resistance are mediated by CD133 signaling to the PI3K pathway	Jamal et al., 2020 [8]
Inhibition of melanoma growth in an autophagy-dependent manner through inhibition of PI3K/AKT/mTOR signaling	Gong et al., 2020 [135]
Inhibition of the PI3K/AKT/mTOR pathway can efficiently counteract dabrafenib-induced stimulation of the invasive capacity of melanoma cells with required resistance.	Caporali et al., 2014 [133]

**Table 2 cells-13-00240-t002:** Studies focused on the reliability of Fyn/CD133/PI3K/mTOR pathway as therapeutic target in melanoma treatment.

Description of the Study	Therapeutic Target	References
Phase II study of the Src kinase inhibitor saracatinib (AZD0530) in metastatic melanoma	NRTK, Fyn	Tang et al., 2020; Gangadhar, et al., 2013 [257,258]
Clinical reliability of PI3K pathway as therapeutic target for melanoma treatment	PI3K	Amaral et al., 2020; Schneider et al., 2014; Tran et al., 2021 [259,260,261]
Clinical relevance of AKT as therapeutic in melanoma treatment	AKT	Ernst et al., 2005; Rebecca et al., 2014 [262,263]
Target both mTORC1 and mTORC2 based on their clinical reliability in melanoma treatment	mTORC1/mTORC2	Sznol et al., 2013; Bernard et al., 2021; Slingluff et al., 2013; Gopal et al., 2014; Espona-Fiedler et al., 2012 [264,265,266,267,268]

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
