# Peer review of "CD133-Dependent Activation of Phosphoinositide 3-Kinase /AKT/Mammalian Target of Rapamycin Signaling in Melanoma Progression and Drug Resistance"

_cells, 2024, doi:10.3390/cells13030240_

Round 1

Reviewer 1 Report (Previous Reviewer 2)

Comments and Suggestions for Authors

Would be great if you could add colors in Figures 4,5, 8-10 

Author Response

Editor

Cells

Dear Editor,

Thank you very much for the encouraging comment regarding our Manuscript ID: Icells-2771771; “CD133-dependent Activation of PI3K PI3K/AKT/mTOR Signaling in Melanoma Progression and Drug resistance.”

As requested, we accordingly modified the information associated with figures 8, 9, 10 based on your comment. First, we checked the references, then we checked the text associated with figures. Also, made some modifications in the figures and accordingly modified the corresponding legends to figures. Now we hope that the manuscript is improved.

Many thanks for your consideration.

On behalf all my co-authors

Authors ‘response to Reviewer 1

Comment: Would be great if you could add colors in Figures 4,5, 8-10

Authors’response: Thank you very much for your valuable comment. Accordingly, we presented the figures in color.

See figures in the whole manuscript.

Reviewer 2 Report (Previous Reviewer 3)

Comments and Suggestions for Authors

Review of

CD133-dependent Activation of PI3K/AKT/mTOR Signaling in Melanoma Progression and Drug Resistance. 

 Overall comments:

This is a resubmission of the manuscript from September ‘23

The main issue was that this review is not specifically reviews the role of PI3K/AKT/mTOR in melanoma but instead in cancer in general.

The authors revised the manuscript somewhat adding many more melanoma-related references.

Unfortunately, they did not update the figures to make them more melanoma-specific. This was a main point in my previous review.

There is no transition as to why the authors mentioning Xiphoporus as a melanoma model on line 277. This needs an introduction. Additionally, the Xiphoporus paragraph (lines 277 – 289) fits more to the “Non-receptor tyrosine kinase Fyn” chapter 6

I don’t understand why some parts are highlighted. They don’t always correspond with new additions to the manuscript.

Please correct / clarify the points below:

Line 69: correct non-genetic

Figure 1:

Please correct interensic -> intrinsic

Figure 2

This figure has not changed, my comments from the last review have not been addressed:

This figure describes the general tumor plasticity, nothing about melanoma.

Differentiation and de-differentiation arrows are in the wrong direction.

As examples, take a look at this publication: https://doi.org/10.3390/ijms161226207 , here: https://doi.org/10.1242/dev.106567 or here: https://doi.org/10.1016/j.ejcb.2013.11.006

Chapter 3, Figure 3:

This figure has not changed, my comments from the last review have not been addressed:

The title is “Melanoma Stem cells” but it is just a general description of cancer stem cells. The melanocyte lineage should be reviewed in this chapter and figure. 

Please have a look at he references I mentioned for Fig 2

Line 277: It’s Xiphoporus.

Figure 6 legend. This has not been corrected:

It is Abl kinase with an “L”

Table 1 is not mentioned in the text. Please highlight its relevance.

Figure 8: This figure has not changed, my comments from the last review have not been addressed:

Figure 8 is not melanoma-specific but it is the general signal transduction pathways downstream of RTKs. Simplifying it according to the text highlighting only melanoma-specific pathways would greatly increase the readability of this figure.

Look at your reference 249 as an example: https://doi.org/10.1038/s41392-021-00827-6

Figure 9: This figure has not changed, my comments from the last review have not been addressed:

Many of these drugs are not used in melanoma treatment but are inhibitors used in general cancer treatments. Just like in Figure 8, simplifying this figure would make it easier to follow, like in reference 249.

  After the authors revising figures 8-10   Looking at these figures closer I feel stronger about that there is no real focus on the role of CD133 on melanoma. Fig8
It's just showing the common pathways in cancer in general.  Recapping reference 249 https://doi.org/10.1038/s41392-021-00827-6  

Fig 9 is repeating figure 8, I don't see the relevance of showing this figure.

Fig 10 hasn't been changed. There should be a reference for each drug and/or target molecule.

Comments on the Quality of English Language

 Minor editing of English language required

Author Response

Overall comments:

This is a resubmission of the manuscript from September ‘23

The main issue was that this review is not specifically reviews the role of PI3K/AKT/mTOR in melanoma but instead in cancer in general.

The authors revised the manuscript somewhat adding many more melanoma-related references.

Comment: Unfortunately, they did not update the figures to make them more melanoma-specific. This was a main point in my previous review.

There is no transition as to why the authors mentioning Xiphoporus as a melanoma model on line 277. This needs an introduction. Additionally, the Xiphoporus paragraph (lines 277 – 289) fits more to the “Non-receptor tyrosine kinase Fyn” chapter 6

I don’t understand why some parts are highlighted. They don’t always correspond with new additions to the manuscript.

Authors response: In the pervious comment of the reviewer. The reviewer asked to add the Xinphorus papragraph to the section of non-receptor tyrosine kinase Fyn and we have already did it

as required.

See the section of non-receptor tyrosine kinase Fyn, Lines 360-383;  the xiphoporus paragraph.

has been added as following [As widely documented, the Src family kinase/Focal Adhesion Kinase (FAK) complex is a signaling platform that is known to play a crucial role in the regulation of oncogenic growth factor receptors-dependent downstream pathways [191, 192]. This observation is more described in the case of melanoma in Xiphophorus fish, in which the oncogenic EGF receptor orthologue Xiphophorus melanoma receptor kinase (Xmrk) effects the continuous activation of the Src family kinase Fyn that is strongly involved in promoting many tumorigenic events [192, 194].

The Xiphophorus fish that is derived from the crosses between X. maculatus (the southern platyfish) and X. hellerii (the green swordtail) species of the fish genus Xiphophorus [193, 194],  can spontaneously develop malignant melanoma via a proto-oncogene encoding a receptor tyrosine kinase designated Xmrk-dependent mechanism [193]. The encoded Xmrk protein is structurally related to the human EGFR with an extracellular ligand-binding domain, a transmembrane domain and an intracellular

The Melanoma formation in Xiphoporus is initiated by overexpression of the EGFR-related receptor tyrosine kinase Xmrk. This receptor is activated in fish melanoma as well as in a melanoma-derived cell line (PSM) resulting in constitutive Xmrk-mediated mitogenic signaling. The elevated expression of Xmrk is the initial cell-type specific event in melanoma formation in Xiphoporus [110]. Xmrk-mediated transformation po-tential is cell-type specific signal transduction-dependent mechanism [195]. The NRTK Fyn has been identified as substrate of Xmrk in Xiphoporus melanoma cells [183, 196]. In addition, Xmrk has been reported to contains binding sites for growth factor receptor bound protein 2 (GRB2), Src homology and Collagen (Shc), Fyn and PLCg [198]. Also, the binding of Grb2 to the activated Xmrk has been shown to triger activation of the MAP kinase pathway [196]. While the binding of Fyn to the Xmrk receptor is mediated through its SH2 domain [197].

FYN is highly expressed in many cancers and promotes cancer growth and metastasis through diverse biological functions such as cell growth, apoptosis, and motility migration, as well as the development of drug resistance in many tumors [12, 153, 179-183]. In addition, FYN is involved in the regulation of multiple cancer-related signaling pathways, including interactions with ERK, COX-2, STAT5, MET and AKT [198].  FYN is therefore an attractive therapeutic target for various tumor types, and suppressing FYN can improve the prognosis and prolong the life of patients.] has been added.

Please correct / clarify the points below:

Comment: Line 69: correct non-genetic

Authors’respone: We corrected it.

See line:64. The word non-gegitic  is corrected to non-genetic

Comment: Figure 1:

Please correct interensic -> intrinsic

Authors’response: thank you for your observation. Accordingly, the word interensic is corrected to become intrinsic: See Figure 1

Comment: Figure 2

This figure has not changed, my comments from the last review have not been addressed:

This figure describes the general tumor plasticity, nothing about melanoma.

Differentiation and de-differentiation arrows are in the wrong direction.

As examples, take a look at this publication: https://doi.org/10.3390/ijms161226207 , here: https://doi.org/10.1242/dev.106567 or here: https://doi.org/10.1016/j.ejcb.2013.11.006

Authors’ response: We have replaced the current figure with a new one. In addition, we modified this paragraph to be more melanoma specific.

See the lines: 169-189; the following text  [Melanoma cells have the potential to switch their phenotype during tumor progression, from a proliferative and differentiated phenotype to a more invasive and dedifferentiated phenotype. [79, 80. Genetic alterations, influences from the tumor microenvironment and epigenetic changes belong to the phenotypic plasticity and high heterogeneity that is known to be characteristic for melanoma [81.]. Melanoma cells continuously undergo reversible alteration between a proliferative/differentiated and an invasive/dedifferentiated phenotype, an epithelial-to-mesenchymal transition (EMT)-like process [82, 83]. Thus, the transition of melanoma cells into an invasive phenotype facilitates melanoma dissemination from a primary tumor to distant sites [84, 8585]. However, phenotype alteration is mediated mainly by a hypoxic tumor microenvironment and inflammatory signals-dependent mechanisms [86-89].

In addition to the reversible alteration between a proliferative/differentiated and an invasive/dedifferentiated phenotype [79, 90, 91], occurrence of melanoma cell plasticity is mediated by both cell-autonomous mechanisms and tumor microenvironment-dependent signals [90, 92]. Accordingly, cancer cell/stromal cell-dependent mechanisms can impact the regulation of the phenotypic plasticity of melanoma cells [90-95]. For example, cancer associated fibroblasts (CAFs), the major component of the tumor microenvironment have been shown to , play key roles in the regulation of melanoma cell plasticity via hepatocyte growth factors (HGF)-dependent mechanism [96, 97] and insulin-like growth factor receptor signaling -dependent mechanisms [98]. .

Thus, understanding the biological complexity of cancer cell plasticity and its role in melanoma progression and relapse may lead to the development of new therapeutic approaches for the treatment of melanoma. Also, the functional analysis of specific cell markers and key molecules of aberrant signaling pathways, particularly those closely associated with the maintenance of stemness properties and the regulation of the cross talk between melanoma cells and their microenvironment, is urgently needed. The mechanisms regulating melanoma plasticity are outlined in Figure 2.].

References

Comment: Chapter 3, Figure 3:

This figure has not changed, my comments from the last review have not been addressed:

The title is “Melanoma Stem cells” but it is just a general description of cancer stem cells. The melanocyte lineage should be reviewed in this chapter and figure. 

Please have a look at the references I mentioned for Fig 2

Authors response: We changed the figure and made more specific for melanoma stem cells. In addition, we modified the text and legend of the figure.

See lines: 197-212; and lines 214-229.

The following text [ 3. Melanoma stem cells. The processes of melanoma development start in mature melanocytes [99, 100]. Although accumulated evidence indicates that early-stage precursors of melanocytes exist in the dermis [101, 102], other reports suggested that earliest origins of cutaneous melanoma may have occurred in extrafollicular melanocyte stem cells [100-102]. It is probably that melanoma stem cells (MSCs) are derived either from a transformed melanocyte, from a transformed melanocyte stem cell, or from a combination of both sources. Like CSCs, MSCs are characterized by their ability to self-renew and differentiate [103, 104, 105]. The generation of these subpopulations is mediated via genetic segregation and epigenetic alterations via transcriptional regulation of genes associated with stemness properties and dysregulation of aberrant signaling pathways such as FOXM1 signaling [106-108]. Consequently, melanoma cells undergo intrinsically asymmetric cell divisions of stem cell lineage to produce two daughter cells, both significantly different in their genetic material and phenotype, [109-111]. One of these daughter cells is characterized by its stemness properties and is referred to as CSC, whereas the other daughter cell (non-CSC) represents a larger portion of the overall tumor mass [4,5,9,, 110, 111. In addition to the deregulation of aberrant signaling pathways for tumor growth and survival, MSCs are notable for their expression of CD20, CD133, CD166, CD271, ABCB5, Nestin and CD105 surface markers [4,5, 8]. Thus, understanding the pathways controlling self-renewal, expansion, and differentiation of MSCs and how UV radiation alters and disrupts melanocyte lineage pathways will bring greater insight into the origins of melanomas. The development of CSCs from normal stem/progenitor cells (Fig. 3A) and from cancer cells (Fig.3B) is outlined in detail. ]

Legend to Fig. 3;  See lines: 214-222 [ Figure 3. Generation of cancer stem cells (CSCs) from both tumor and normal cells. A) Generation of cancer stem cells (CSCs) from cancer cells is mediated by the activation of aberrant signaling pathways via driver mutation-dependent mechanisms and mechanisms that turn on self-renewal genes. Cancer cells become programmed to divide into cancer progenitor cells and CSCs. Cancer progenitor cells are genetically programmed to divide into one differentiated cell and one CSC. Once the dedifferentiation process of the differentiated cell has been completed, it can be transformed into a CSC that ultimately undergoes abnormal asymmetric cell division to produce genetically divergent subpopulations including CSCs and non-CSCs. B) The development of CSCs from normal stem/progenitor cells is mediated by multiple genetic mutations and dedifferentiation-dependent mechanisms, whereby normal stem/progenitor cells undergo abnormal asymmetric cell division to produce two genetically divergent subpopulations including CSCs and non-CSCs.]

Beyond their identification in various malignancies as part of the tumor mass, CSCs are characterized by their ability to confer self-renewal, differentiation, tumor initiation, metastasis, recurrence, and drug resistance [5, 6, 8, 112]. Like CSCs of different tumor types, MSCs have been functionally characterized in vitro and in vivo [5, 6, 9,100, 102]. In addition to their stemness properties, MSCs demonstrate the expression of stem cell markers including CD20, CD105, CD133, CD146, CD166, CD271, ABCB5, and Nestin [5]. Apart from their function as stem cell marker proteins, such as CD133 protein have been discussed for their functional role in the regulation of MSCs maintenance and resistance [8]. ] has been modified .

Comment: Line 277: It’s Xiphoporus.

Authors’response: We corrected it. See lines:362, 369, 371 and 373

Comment: Figure 6 legend. This has not been corrected:

It is Abl kinase with an “L”

Authors Response: Thank you very much for your response. We corrected it; See Lines: 309, 310, 319, 335

Comment: Table 1 is not mentioned in the text. Please highlight its relevance.

Authors’response: Tab. 1 is mentioned in the text. See line: 284

Comment: Figure 8: This figure has not changed, my comments from the last review have not been addressed:

Figure 8 is not melanoma-specific but it is the general signal transduction pathways downstream of RTKs. Simplifying it according to the text highlighting only melanoma-specific pathways would greatly increase the readability of this figure.

Look at your reference 249 as an example: https://doi.org/10.1038/s41392-021-00827-6

Authors response: Thank you very much for your comment. Accordingly, modifed the figure and made it more simpler. See Figure 8

Authors’response:  Fig. 8 has been simplified and contains two parts. A part describe the Non-receptor tyrosine kinase mediated signaling pathways in melanoma. The outline of this figure based on the data reported in melanoma. While the other part, Fig. B outlines Receptor tyrosine kinase induced pathways in melanoma based on the reported studies in melanoma. Thus, Fig. has been proposed compare between non-receptor tyrosine kinase Fyn- stimulated CD133 signal leading to the activation of PI3K/AKT that is expected to enhance the activity mTORC1 and Receptor tyrosine kinase-induced activation of PI3K/AKT/mTORC1

Comment: Figure 9: This figure has not changed, my comments from the last review have not been addressed:

Many of these drugs are not used in melanoma treatment but are inhibitors used in general cancer treatments. Just like in Figure 8, simplifying this figure would make it easier to follow, like in reference 249.

Authors’response: We modified the figure as required. In this figure there no inhibitors are mentioned. We just demonstrated the possible pathways , which are regulated Receptor tyrosine kinase -dependent activation

Comment:  After the authors revising figures 8-10   Looking at these figures closer I feel stronger about that there is no real focus on the role of CD133 on melanoma. Fig8
It's just showing the common pathways in cancer in general.  Recapping reference 249 https://doi.org/10.1038/s41392-021-00827-6  

Authors ‘response: Thank you very much for your comment. We and others have already demonstrated the role of CD133 signal in the activation of PI3K and its downstream pathways. Thus, based on our reports and reports of others. There is a direct link between Fyn-stimulated CD133 signal to PI3K/AKT and the activation of mTOR in melanoma and in other tumor types. We characterized The CD133 signal to PI3K/AKT in melanoma Jama et al., 2020 and We et al. demonstrated the CD133 signal to PI3K/AKT in glioblastoma and Boivin et al., demonstrated the phosphorylation of the tyrosine residues located on the cytoplasmic domain of CD133.  Thus, based on these findings, we address a role for Fyn/CD133/PI3K/AKT in the activation of mTOR in melanoma. In addition, we modified this paragraph and highlighted the role of CD133 in the context of the activation mTOR in melanoma.

Comment: Fig 9 is repeating figure 8, I don't see the relevance of showing this figure.

Authors’response: We generated this Fig. 9 to outline in detail receptor tyrosine kinases-induced pathways in melanoma.

Comment: Fig 10 hasn't been changed. There should be a reference for each drug and/or target molecule.

Authors’ response: We modified Fig.10 and, modified the corresponding text and made it more specific for melanoma.

See Lines: 527-536; the following text [Many of the multiple targeted inhibitors for Fyn/CD133, PI3K, AKT, or mTOR have been evaluated and tested in vitro melanoma model and in phase I and II clinical trials in melanoma. These include the inhibition of Fyn/Stat pathway by chalcone derivative in melanoma [259]. The inhibitor of Fyn/CD133 (Saracatinib) that has been evaluated by Phase II Study in Metastatic Melanoma [260] and the inhibitors of PI3K (BAY-80-6946) [261] and of buparlisib [262] , the inhibitors of AKT including perifosine in patients with metastatic melanoma [263] and MK-2206 in BRAF wild type melanoma [264]. While the dual inhibitor of PI3K/mTOR (NVP-BEZ235) has been studied in melanoma [265]. Also, the mTOR analogs include everolimus [266] and temsirolimus [267] have been tested up on their inhibitory effect on mTORC1 in melanoma; conversely, mTOR kinase inhibitors were found to target both mTORC1 and mTORC2 [268, 269]. In addition, the most common references describing the reliability of Fyn/CD133/PI3K/mTOR pathway as therapeutic target in melanoma treatment have been demonstrated in table 2. While the possible therapeutic targets of the Fyn/CD133/PI3K/Akt/mTOR in melanoma are outlined in figure.10. ] has been modified  

In addition, we proved the references in the whole manuscript and cited the most related reference to melanoma and the references that may support our hypothesis.

Round 2

Reviewer 2 Report (Previous Reviewer 3)

Comments and Suggestions for Authors

The authors claim to have made the manuscript somewhat more melanoma-specific.

The highlighted lines do not correspond with changes in the manuscript. This is very confusing. The authors claim to have made changes to parts of the manuscript but it still is unchanged.

Figure 2 is improved, but legend is still the same although authors state they have changed it

Figure 3 and its legend is still the same although the authors claim they have changed it.

Table 1 is still not mentioned (line 284 would be in the Fig. 4 legend)

Figure 8 hasn't been simplified, rather made more complicated.

Figure 9 is still the same although the authors claim they have changed it. It's now in color. What do the different colored arrows mean?

The authors write in the rebuttal to fig. 9 this:

" ... In addition, we modified this paragraph and highlighted the role of CD133 in the context of the activation mTOR in melanoma."

Which paragraph has been changed?

. Added an intro for

Comments on the Quality of English Language

still some typos

Author Response

Editor in Chief of Cells,

Dear Editor,

Thank you very much for the encouraging comment. We answered point for point, the reviewer 3’s comment. By the changing of legends to figure 2 and 3. In addition, the modification, and the improvement of figures 8, 9 and 10 have been made in the former response by the first revision. According to the former modification and changes, we feel both Figures 8 and 9 are easy for the reader to follow. Also, colored arrows were done to distinguish between the different outcomes induced RTKs-mediated PI3K/AKT. Enclosed please find our response to the valuable comment of Reviewer 3.

On behalf all my coauthors

Authors’ response to Comment Reviewer 3

The authors claim to have made the manuscript somewhat more melanoma specific.

The highlighted lines do not correspond with changes in the manuscript. This is very confusing. The authors claim to have made changes to parts of the manuscript, but it still is unchanged.

Comment: Figure 2 is improved, but legend is still the same although authors state they have changed it

Authors’ response: Thank you very much for your comment. Accordingly, we modified the legend to figure 2.

See lines 217-224: the following the current legend to figure [ Figure 2. Mechanisms of melanoma plasticity. Melanoma cell plasticity is the ability of tumor cells to shift dynamically between differentiated/cancer stem cells (CSCs) and undifferentiated /non-cancer stem cells (non-CSCs) states to promote long-term tumor cell growth, invasiveness, and migration. The mechanisms of melanoma plasticity are mediated by genetic and epigenetic modifications of melanoma cells and by significant alteration of tumor microenvironmental secretory products (e.g., growth factors and cytokines), activation of cancer-associated fibroblasts (CAFs) or tumor-associated macrophages (TAMs).] has been replaced by a new one [ Figure 2. Cellular plasticity in melanoma. Melanoma cells can switch between a differentiated/proliferative/cancer stem cells (CSCs) and a dedifferentiated/invasive phenotype /non-stem cancer cells via mechanism mediated by genetic and epigenetic alterations, and via crosstalk between melanoma cells and their microenvironment as well. Cancer cells can be reprogrammed towards pluripotency. Phenotype switch of melanoma increases their plasticity and is responsible for tumor growth, invasion and drug resistance. ].

Comment: Figure 3 and its legend is still the same although the authors claim they have changed it.

Authors’ response: Thank you very much for your comment. As required, we changed the legend to figure 3.

See lines: 249-260   The following legend [Figure 3. Proposed model for cancer stem-like cells (CSCs)/melanoma stem cells (MSCs) generation from normal and cancer cells. A) Generation of CSCs/MSCs results from the transformation of normal stem/ progenitor cells into undifferentiated cancer cells via multiple genetic mutations and dedifferentiation-dependent mechanisms. B) Generation of MSCs from cancer cells, this model describes the generation of MSCs from tumor cells through the activation of aberrant signaling pathways via driver mutations to tumor growth signaling pathways and turning on of self-renewal genes. MSCs become educated to divide into tumor progenitor cells/MSCs. The produced MSCs possess genetic properties which allow for the division into one differentiated cell and one MSC. Once the dedifferentiation process of the differentiated cell is complete, the dedifferentiated cells can be transformed into MSCs. CSCs/MSCs undergo abnormal asymmetric cell division to produce two daughter cells one as CSC and one as non-CSCs.] has been modified and replaced by this legend [Figure 3. Model for the development of cancer stem cells (CSCs) from both tumor and normal cells. A) Development of CSCs from cancer cells is mediated by the activation of aberrant signaling pathways via driver mutation-dependent mechanisms and transcriptional activation of self-renewal genes. Cancer cells become programmed to divide into cancer progenitor cells and CSCs. Cancer progenitor cells are genetically programmed to divide into one differentiated cell and one CSC. Once the dedifferentiation process of the differentiated cell has been completed, it can be transformed into a CSC that ultimately undergoes abnormal asymmetric cell division to produce genetically divergent subpopulations including CSCs and non-CSCs. B) The development of CSCs from the activation of self-renewal genes normal stem/progenitor cells via multiple genetic mutations and dedifferentiation-dependent mechanisms.  Like cancer cells derived CSCs, CSCs derived from normal stem/progenitor cells undergo abnormal asymmetric cell division to produce two genetically divergent subpopulations including CSC and non-CSC. ].

Comment:  Table 1 is still not mentioned (line 284 would be in the Fig. 4 legend)

Authors’ response: Thank you very much for your comment. We mentioned the table in the text and corrected the legend to Fig. 4

See legend to Fig. 4

See lines: 336-337; the following sentence [In addition, the references describing functional role of Fyn-stimulated CD133 signal in melanoma has been mentioned (Tab.1).] has mentioned in the text of the manuscript.

Comment: Figure 8 hasn't been simplified, rather made more complicated.

Authors’ response: Thank you very for your comment. We discussed RTKs and NRTKs -pathways to melanoma progression and drug resistance. In addition, we have drawn Fig. 8 to make it easier to understand. We see that Fig. 8 reflects the data that has been already reported on the mechanisms of RTKs and RTKs-mediated pathways to melanoma progression.

Comment: Figure 9 is still the same although the authors claim they have changed it. It's now in color. What do the different colored arrows mean?

Authors’ response: Thank you very much for your comment. We have already changed Fig.9 in the first revision. Please compare the first submission of the manuscript and the resubmitted first revision. We have used different colored arrows to make it easier for the reader to follow the outcome of RTKs-mediated activation of PI3/AKT in melanoma.

Comment: The authors write in the rebuttal to fig. 9 this:

" ... In addition, we modified this paragraph and highlighted the role of CD133 in the context of the activation mTOR in melanoma."

Which paragraph has been changed?

Authors’response: Thank you very much for your comment. We have already done the modification of the paragraph focusing on the role of CD133 signal in the context of mTOR by the first revision. Please compare the content of this paragraph in the first submission and the content of the same paragraph in the current revision.

See lines:553-571; the following paragraph [The target of rapamycin (TOR) protein was first identified in the budding yeast Saccharomyces cerevisiae, as a target of rapamycin [138, 139]. While the structurally and functionally conserved mammalian counterpart mTOR was identified based on its biochemically and inhibitory properties to rapamycin in mammalian cells [140].

mTOR includes two functionally distinct protein complexes, namely mTOR complex 1 (mTORC1) and mTOR complex 2 (mTORC2) [141]. mTORC1 is composed of mTOR, raptor, mLST8, and two negative regulators, PRAS40 and DEPTOR [142]. While mTORC2 is composed of the conserved mTOR, RICTOR (mAVO3), SIN1 and mLST8 (GβL) as well as the less-conserved proteins such as PRR5/Proctor, PRR5L and DEPTOR [143].

The activation of mTORC1 is mediated mainly by PI3K/AKT pathway, while its inhibition is regulated by the major regulator of ribosomal biogenesis and protein synthesis, TSC1/TSC2 complex via the activation of S6K and inactivation of the repressor of mRNA translation-dependent mechanism (4EBP1) [144]. Apart from the mechanisms regulating mTORC1 activation and inhibition, the involvement of mTORC1 in the dysregulation of many functional proteins has been reported. These function proteins, including cytoplasm linker protein-170 (CLIP-170), eukaryotic elongation factor 2 (eEF2) kinase, ornithine decarboxylase (ODC), glycogen synthase, hypoxia-inducible factor 1α (HIF-11α), lipin, PKCδ and PKCε, protein phosphatase 2A (PP2A), cyclin-dependent kinase inhibitors, p21Cip1 and p27Kip1, retinoblastoma (Rb) protein, and signal transducer and activator of transcription (STAT-3) [52, 145,146]. While the important role of mTORC2 in the promotion of cancer cell survival, proliferation, growth, and motility is attributed to its ability to enhance the phosphorylation of AktSer473, the key regulator of insulin/PI3K pathway [147-149]. Also, the permanent activation of AKT has been reported to trigger feedback reactivation of mTORC2 [150, 151]. Thus, targeting mTORC1 or mTORC2 may have a therapeutic impact on the treatment of different tumor types including melanoma. Fig. 9 outlines the mechanisms of RTKs and NRTKs-mediated signal to PI3K/AKT/mTOR pathway in melanoma cells.] has been modified and replaced by the following paragraph [The activation of mTORC1 is mediated by the inactivation of both TSC1 and TSC2 following their phosphorylation  by the PI3K/AKT pathway [207]. In addition to its functional role in the  regulation of cell growth, proliferation and survival in response to sensing mitogen, energy and nutrient signals, the mTORC1 is involved in the regulation of many functional proteins including the regulation of eukaryotic elongation factor 2 (eEF2) kinase [214, 215], CLIP-170 (cytoplasmic linker protein-170) [216], ornithine decarboxylase (ODC)  [217], hypoxia-inducible factor 1α (HIF-1α) [218], protein phosphatase 2A (PP2A) [219] lipin [220], PKCδ and PKCε [221] [225], protein phosphatase 2A (PP2A)[2], p21Cip1and p27Kip1cyclin-dependent kinase inhibitors [222, 223] and  retinoblastoma protein (Rb) [224].

While the second complex of mTOR, namely mTORC2, has been demonstrated to play an  important role in the promotion of cancer cell survival, proliferation, growth, and motility based on  its ability to enhance the phosphorylation of AktSer473, the key regulator of insulin/PI3K pathway [9, 225]. To that end, the permanent activation of AKT is associated with feedback reactivation of mTORC2 [226]. Thus, targeting mTORC1 and/or mTORC2 may have a therapeutic impact on the treatment of different tumor types, including melanoma. The mechanisms of RTKs -mediated signaling to PI3K/AKT/mTOR pathway its biological consequences in melanoma cells is outlined in figure 9.]

This manuscript is a resubmission of an earlier submission. The following is a list of the peer review reports and author responses from that submission.

Round 1

Reviewer 1 Report

Comments and Suggestions for Authors

The review article entitled ‘CD133-dependent Activation of PI3K/AKT/mTOR Signaling in Melanoma Progression and Treatment Resistance’ was well received. In this article the authors have described the role of CD133-mediated PI3K PI3K/AKT/mTOR signaling in melanoma progression.

In the abstract section the authors have stated and I quote ‘Herein, the role of CD133-dependent activation of PI3K/mTOR in the regulation melanoma progression, treatment resistance, and recurrence is reviewed. But, the role of CD133-dependent activation of P13K/mTOR in (1) MELANOMA PROGRESSION (2) TREATMENT RESISTANCE (3) RECURRANCE is missing or scarce in current study. Please add specific information.

A table containing different studies focused on CD133 dependent P13K/AKT/Mtor activation in melanoma progression should be added.

The information in the introduction section does not contain anything about CD133. A paragraph about the role of CD133 in introduction section will help reader get an overview of the study at the beginning. The authors can give a brief introduction about the role of CD133 in PI3K/AKT/mTOR pathway and or Melanoma.

Comments on the Quality of English Language

Fine

Author Response

Editor

Cells

Dear Editor,

Thank you very much for the encouraging comment regarding our Manuscript ID: ijms-2676058; “CD133-dependent Activation of PI3K PI3K/AKT/mTOR Signaling in Melanoma Progression and Drug resistance.”

As required, enclosed find please our response “Point-for-Point” to the valuable comments of

Reviewer1

Comments and Suggestions for Authors

The review article entitled ‘CD133-dependent Activation of PI3K/AKT/mTOR Signaling in Melanoma Progression and Treatment Resistance’ was well received. In this article the authors have described the role of CD133-mediated PI3K PI3K/AKT/mTOR signaling in melanoma progression.

Comment: In the abstract section the authors have stated, and I quote ‘Herein, the role of CD133-dependent activation of PI3K/mTOR in the regulation melanoma progression, treatment resistance, and recurrence is reviewed. But the role of CD133-dependent activation of P13K/mTOR in (1) MELANOMA PROGRESSION (2) TREATMENT RESISTANCE (3) RECURRANCE is missing or scarce in current study. Please add specific information.

Authors’ response: Thank you very much for your valuable comment. Accordingly, we added more information about the role of CD133-dependent activation of PI3K/mTOR in treatment resistance and recurrence.

See lines: 252-287; the following paragraphs [CD133-expressing CSCs have been shown to exhibit resistance to chemotherapy and radiation therapy in addition to be associated with poor prognosis in various cancers [120-123]. We and others demonstrated that CD133+ cancer cells confer resistance to many chemotherapeutic agents such as Caffeic acid phenethyl ester [5], Taxol [6], and fotemustine [9]. Accordingly, CD133-dependent mechanisms have been shown to be involved in the development of melanoma resistance to chemotherapy [9].

The contribution of CD133 to the regulation of CSCs functions such as self-renewal, differentiation, and drug resistance are likely mediated by the NRTK, Fyn-dependent mechanism via the phosphorylation of Tyr828 residue located on the cytoplasmic domain of CD133 [124-126]. Our laboratory has demonstrated that the phosphorylation of Tyr828 is essential to trigger the activation of PI3K and its downstream dependent signaling pathways in melanoma [9].

The PI3K/AKT pathway is one of the most important networks with the highest mutation frequency in human cancers [125]. Both PI3K/AKT/NF-κB and PI3K/AKT/ mTOR are the two main mutated pathways involved in apoptosis and tumorigenesis facilitating the development of tumor resistance to anti-cancer agents [127]. Dysregulation of major key molecules of these signaling pathways is associated with drug resistance and melanoma progression. Also, elevated activation of PI3K pathway has been suggested to trigger melanoma progression by the activation of PI3K/AKT/ NF-κB axis. [128-130].

In addition to the key role of CD133 in chemoresistance, we and others have demonstrated the cellular mechanisms by which the CD133 protein triggers activation of PI3K pathway both in melanoma and glioblastoma [9, 131].

The PI3K/AKT/mTOR signal to downstream proteins leades to the development of tumor resistance [132, 133]. Also, the PI3K/AKT/mTOR pathway has been shown to play a crucial role in a variety of biological and physiological processes including survival, growth, transcription, and translation, which are associated with the development of drug resistance [134, 135]. Abnormal activation of the PI3K/AKT/mTOR pathway in different tumor types including melanoma has been suggested to be the key mechanism through which tumors evade drug toxicity [136, 137]. Thus, CD133 -mediated activation of PI3K/AKT pathway in melanoma [9] and glioma [138] cells and contributes to the activation of mTORC1 through the phosphorylation of mTOR at Ser2448.

In addition to the frequent mutation to AKT family members, mutations to the PI3K-AKT-mTOR pathway are common in melanoma [139, 140]. Mutation in PTEN has been shown to effectively restrain PI3K/AKT/mTOR growth-promoting signaling cascade in primary and metastatic melanoma patients [141, 142]. Upon activation of PI3K or AKT the expression of mTORC1 increases, while its subsequent activation of mTORC1 results in the phosphorylation of the downstream molecules, p70S6K1, and eukaryotic initiation factor 4EBP1, which, in turn, affect mRNA translation and protein synthesis [140-143]. Thus, the key mechanism for the regulation of both PI3K/AKT and PI3K/NF-κB pathways is mediated by Fyn-stimulated CD133 signaling to PI3K [125, 144-146]. To that end, Fyn-stimulated CD133 signaling to PI3K is responsible for the activation of the mTOR allowing melanoma progression [147, 148] drug resistance [149] and recurrence [150]. The mechanisms through which Fyn-stimulated CD133 signal leading to the activation of to PI3K and its downstream pathways are outlined in detail (Fig. 5).] have been edited.

Comment: A table containing different studies focused on CD133 dependent P13K/AKT/Mtor activation in melanoma progression should be added.

Authors’ response: Thank you very much for your comment. Although the CD133-dependent activation of PI3K in melanoma progression has been reported in limited studies. The activation of PI3K/AKT/mTOR pathway has been demonstrated in several studies. However, we and others suggested CD133-dependent activation of PI3K in melanoma and Glioma cells. Also, the phosphorylation of Tyrosine 828 and Tyrosine 852 residues located on the cytoplasmic domain of CD133 Fyn leading to the activation of PI3K to generate the following PI3K/AKT, PI3K/mTOR and PI3K/NF-KB pathways. Accordingly, the following changes has been made in the manuscript.

See Tab. 1; the following Table [Table 1. Studies focused on CD133- dependent P13K/AKT pathway signaling and its link to mTOR activation in melanoma progression.

Description of the study

References

Melanoma progression and treatment resistance are mediated by CD133 signaling to the PI3K pathway

Jamal et al., 2020 [9]

Inhibition of melanoma growth in an autophagy-dependent manner through inhibition of PI3K/AKT/mTOR signaling

Gong et al., 2020 [139]

Inhibition of the PI3K/AKT/mTOR pathway can efficiently counteract dabrafenib-induced stimulation of the invasive capacity of melanoma cells with required resistance.

Caporali et al., 2014 [140]

And the following references.

See Lines: 583-585 and 903-907 the following references have been highlighted references section. [ 9. Jamal, S. M. E.; Alamodi, A.; Wahl, R. U.; Grada, Z.; Shareef, M. A.; Hassan, S. Y.; Murad, F.; Hassan, S. L.; Santourlidis, S.; Gomez, C. R.; et al. Melanoma stem cell maintenance and chemo-resistance are mediated by CD133 signal to PI3K-dependent pathways. Oncogene 202039 (32), 5468-5478. DOI: 10.1038/s41388-020-1373-6.

  1. Gong, C.; Xia, H. Resveratrol suppresses melanoma growth by promoting autophagy through inhibiting the PI3K/AKT/mTOR signaling pathway. Exp Ther Med 202019(3), 1878-1886. DOI: 10.3892/etm.2019.8359.

  1. Caporali, S.; Alvino, E.; Lacal, P. M.; Levati, L.; Giurato, G.; Memoli, D.; Caprini, E.; Antonini Cappellini, G. C.; D'Atri, S. Targeting the PI3K/AKT/mTOR pathway overcomes the stimulating effect of dabrafenib on the invasive behavior of melanoma cells with acquired resistance to the BRAF inhibitor. Int J Oncol 201649(3), 1164-1174. DOI: 10.3892/ijo.2016.3594.] have been added to the manuscript

Comment: The information in the introduction section does not contain anything about CD133. A paragraph about the role of CD133 in the introduction section will help reader get an overview of the study at the beginning. The authors can give a brief introduction about the role of CD133 in PI3K/AKT/mTOR pathway and or Melanoma.

Authors’ response: Thank you very much for your comment. Accordingly, we added a paragraph about the role of CD133 and the role of CD133in PI3K/AKT/mTOR in melanoma to the introduction section.

See Lines: 49-60; a paragraph on melanoma and melanoma stem cells [Human malignant melanoma is a highly aggressive skin cancer. characterized by its heterogeneity, and propensity to metastasize to distant organs and the potential for developing resistance to conventional and even the newest targeted therapeutics as measured by progression-free and overall survival [1,2,3]. As heterogeneous tumor, malignant melanoma exists in the form of genetically divergent subpopulations containing melanoma initiating cells/cancer stem-like cells (CSCs) as a small fraction, and non-cancer stem cells (non-CSCs) that form most tumor mass [4-6]. Like other CSCs, melanoma stem-like cells (MSCs) are characterized by their unique surface proteins and aberrant signaling pathways [4-6], which are either in a causal or consequential relationship to melanoma progression, treatment resistance and recurrence [5-9]. CD133 (Prominin-1) is one of the most important cancer stem cells (CSC) that is widely expressed in CSC subpopulation derived from a large variety of human malignancies, including melanoma [5, 6, 9]. Byond its role as a reliable CSC marker for the identification of CSC populations [7, 8], accumulating evidence indicated that CD133 is responsible for CSCs tumorigeneses, metastasis and chemoresistance [4, 8, 9].]  has been added to the introduction section.

See lines: 252-287; a couple of paragraphs on the role of CD133 and the role of CD133 in the activation PI3K/mTOR [CD133-expressing CSCs have been shown to exhibit resistance to chemotherapy and radiation therapy in addition to be associated with poor prognosis in various cancers [120-123]. We and others demonstrated that CD133+ cancer cells confer resistance to many chemotherapeutic agents such as Caffeic acid phenethyl ester [5], Taxol[6], and fotemustine [9]. Accordingly, CD133-dependent mechanisms have been shown to be involved in the development of melanoma resistance to chemotherapy [9].

The contribution of CD133 to the regulation of CSCs functions such as self-renewal, differentiation, and drug resistance are likely mediated by the NRTK, Fyn-dependent mechanism via the phosphorylation of Tyr828 residue located on the cytoplasmic domain of CD133 [124-126]. Our laboratory has demonstrated that the phosphorylation of Tyr828 is essential to trigger the activation of PI3K and its downstream dependent signaling pathways in melanoma [9].

The PI3K/AKT pathway is one of the most important networks with the highest mutation frequency in human cancers [125]. Both PI3K/AKT/NF-κB and PI3K/AKT/ mTOR are the two main mutated pathways involved in apoptosis and tumorigenesis facilitating the development of tumor resistance to anti-cancer agents [127]. Dysregulation of major key molecules of these signaling pathways is associated with drug resistance and melanoma progression. Also, elevated activation of PI3K pathway has been suggested to trigger melanoma progression by the activation of PI3K/AKT/ NF-κB axis. [128-130].

In addition to the key role of CD133 in chemoresistance, we and others have demonstrated the cellular mechanisms by which the CD133 protein triggers activation of PI3K pathway both in melanoma and glioblastoma [9, 131].

The PI3K/AKT/mTOR signal to downstream proteins leades to the development of tumor resistance [132, 133]. Also, the PI3K/AKT/mTOR pathway has been shown to play a crucial role in a variety of biological and physiological processes including survival, growth, transcription, and translation, which are associated with the development of drug resistance [134, 135]. Abnormal activation of the PI3K/AKT/mTOR pathway in different tumor types including melanoma has been suggested to be the key mechanism through which tumors evade drug toxicity [136, 137]. Thus, CD133 -mediated activation of PI3K/AKT pathway in melanoma [9] and glioma [138] cells and contributes to the activation of mTORC1 through the phosphorylation of mTOR at Ser2448.

In addition to the frequent mutation to AKT family members, mutations to the PI3K-AKT-mTOR pathway are common in melanoma [139, 140]. Mutation in PTEN has been shown to effectively restrain PI3K/AKT/mTOR growth-promoting signaling cascade in primary and metastatic melanoma patients [141, 142]. Upon activation of PI3K or AKT the expression of mTORC1 increases, while its subsequent activation of mTORC1 results in the phosphorylation of the downstream molecules, p70S6K1, and eukaryotic initiation factor 4EBP1, which, in turn, affect mRNA translation and protein synthesis [140-143]. Thus, the key mechanism for the regulation of both PI3K/AKT and PI3K/NF-κB pathways is mediated by Fyn-stimulated CD133 signaling to PI3K [125, 144-146]. To that end, Fyn-stimulated CD133 signaling to PI3K is responsible for the activation of the mTOR allowing melanoma progression [147, 148] drug resistance [149] and recurrence [150]. The mechanisms through which Fyn-stimulated CD133 signal leading to the activation of to PI3K and its downstream pathways are outlined in detail (Fig. 5). ] have been added to the text.

See lines:99-103; a paragraph on mTOR [The conserved serine/threonine kinase mTOR, the mammalian target of rapamycin, is a downstream effector of the PI3K/AKT pathway forms two distinct multiprotein complexes, mTORC1 and mTORC2 [10, 11]. mTORC1 is sensitive to rapamycin and activates S6K1 and 4EBP1, both of which are involved in mRNA translation [10, 39]. The activation of mTOR is mediated by diverse stimuli including growth factors, nutrients, energy and stress signals as well as signaling pathways, such as PI3K, MAPK [40, 41].] has been added to the introduction section.

Reviewer 2 Report

Comments and Suggestions for Authors

The authors tried to understand the role of CD133 in the development of melanoma. They specifically reviewed the involvement of the CD133-dependent activation of PI3K/mTOR in melanoma. Although the manuscript is nicely written and presented but I still have some suggestions. 

Terms treatment resistance and drug resistance are used in several places, make it uniform. Better don’t say treatment resistance; it’s too broad; saying drug resistance/ chemotherapy resistance, etc. would be better. 

What do you mean by abnormal activation? 

Some statements are not referenced; please check and provide proper citations. 

Line 254 “Our laboratory, has demonstrated that” better revise. Are you sure from your laboratory? I guess not. Please check the entire manuscript for such omissions. 

Only section 4 says about CD133 but the rest of the manuscript talks about other things, not really aligning with the title. 

How about clinical trials of CD133 in melanoma or other cancers? 

What did you learn from this review? What are key questions do you think worth studying? 

Please write the prospects and challenges of CD133. 

The conclusion should be elaborated, better mentioning something past present, and future of CD133. 

Comments on the Quality of English Language

NA

Author Response

Editor

Cells

Dear Editor,

Thank you very much for the encouraging comment regarding our Manuscript ID: ijms-2676058; “CD133-dependent Activation of PI3K PI3K/AKT/mTOR Signaling in Melanoma Progression and Drug resistance.”

As required, enclosed find please our response “Point-for-Point” to the valuable comments of

Reviewer 2

Comments and Suggestions for Authors

The authors tried to understand the role of CD133 in the development of melanoma. They specifically reviewed the involvement of the CD133-dependent activation of PI3K/mTOR in melanoma. Although the manuscript is nicely written and presented but I still have some suggestions. 

Comment: Terms treatment resistance and drug resistance are used in several places, make it uniform. Better don’t say treatment resistance; it’s too broad; saying drug resistance/ chemotherapy resistance, etc. would be better. 

Authors’ response: Thank you very much for your comment. Accordingly, replaced treatment resistance by drug resistance.

Comment: What do you mean by abnormal activation? 

Authors’ response: Thank you very much for your comment. The meaning of abnormal activation means the activation of cellular signaling pathways by genetic (e.g., mutation in genes encoding for key proteins) and epigenetic (e.g., Promoter methylation of key genes) modification in response to pathophysiological conditions to enhance tumor growth, metastasis, and resistance. Therefore, the abnormal activation of PI3K/AKT pathway in response to genetic and epigenetic alterations and consequently drive melanoma growth, metastasis, and resistance.

Comment: Some statements are not referenced; please check and provide proper citations.

Authors’ response: Thank you very much for your valuable comment. Accordingly, we revised the manuscript as required and cited the referred statements with suitable references.

Comment: Line 254 “Our laboratory, has demonstrated that” better revise. Are you sure from your laboratory? I guess not. Please check the entire manuscript for such omissions. 

Authors’ response: Thank you very much for your comment. We have already demonstrated that the phosphorylation of the Tyr 828 and Try852 located on the cytoplasmic domain of CD133 by the NRTK, Fyn [Boivin, D.; Labbé, D.; Fontaine, N.; Lamy, S.; Beaulieu, E.; Gingras, D.; Béliveau, R. The stem cell marker CD133 (prominin-1) is phosphorylated on cytoplasmic tyrosine-828 and tyrosine-852 by Src and Fyn tyrosine kinases. Biochemistry 2009, 48 (18), 3998-4007. DOI: 10.1021/bi900159d.] is responsible for the activation of PI3K/AKT, PI3/PDK-1/AKT and PI3K/AKT/MDM2 pathways through the interaction of p85 protein with Tyr828 residue located on the cytoplasmic domain of CD133 [Jamal, S. M. E.; Alamodi, A.; Wahl, R. U.; Grada, Z.; Shareef, M. A.; Hassan, S. Y.; Murad, F.; Hassan, S. L.; Santourlidis, S.; Gomez, C. R.; et al. Melanoma stem cell maintenance and chemo-resistance are mediated by CD133 signal to PI3K-dependent pathways. Oncogene 2020, 39 (32), 5468-5478. DOI: 10.1038/s41388-020-1373-6.].in melanoma cells. Also, Wei et al has already demonstrated the same mechanism in glioblastoma cells [ Wei, Y.; Jiang, Y.; Zou, F.; Liu, Y.; Wang, S.; Xu, N.; Xu, W.; Cui, C.; Xing, Y.; Cao, B.; et al. Activation of PI3K/Akt pathway by CD133-p85 interaction promotes tumorigenic capacity of glioma stem cells. Proc Natl Acad Sci U S A 2013, 110 (17), 6829-6834. DOI: 10.1073/pnas.1217002110.]]. Also, all these findings have been cited in the manuscript.

See references, Lines: 583-585, 903-907

Comment: Only section 4 says about CD133, but the rest of the manuscript talks about other things, not really aligning with the title. 

Authors’ response: Thank you very much for your valuable comment. Accordingly, we revised the text of the manuscript and added several paragraphs on CD133 to the manuscript.

See Lines: 56-60; the following paragraph [ CD133 (Prominin-1) is one of the most important cancer stem cells (CSC) that is widely expressed in CSC subpopulation derived from a large variety of human malignancies, including melanoma [5, 6, 9]. Byond its role as a reliable CSC marker for the identification of CSC populations [7, 8], accumulating evidence indicated that CD133 is responsible for CSCs tumorigeneses, metastasis and chemoresistance [4, 8, 9].] has been added to the main text of the manuscript.  

See Lines:224-230; the following paragraph [The stem cell marker CD133 (prominin-1/AC133) has a molecular weight of 120 kDa and is encoded by the PROM1 gene [105, 106]. It is a member of pentaspan transmembrane glycoproteins [94]. CD133 localizes specifically to cellular membranes with an extracellular N-terminal domain, 5-transmembrane domains separating two large glycosylated extracellular loops, two small intracellular loops, and an intracellular C-terminal domain [107]. The involvement of the CD133 protein in the maintenance of melanoma stemness properties and drug resistance is mediated by its C-terminal domain, which contains tyrosine binding sites located on tyrosine 828 (Tyr828) and tyrosine 852 (Tyr852) residues [9, 108]. These two tyrosine residues are phosphorylation targets of the non-receptor tyrosine kinase (NRTK) Fyn [109].] has been added to the main text of the manuscript.

See Lines:249-287; the following paragraph [Many studies have demonstrated that increased CD133 expression is associated with high tumorigenicity and metastatic potential for melanoma cells [117]. Also, CD133 protein has been implicated in the regulation of tumor metastasis [118, 119].

CD133-expressing CSCs have been shown to exhibit resistance to chemotherapy and radiation therapy in addition to being associated with poor prognosis in various cancers [120-123]. We and others demonstrated that CD133+ cancer cells confer resistance to many chemotherapeutic agents such as Caffeic acid phenethyl ester [5], Taxol [6], and fotemustine [9]. Accordingly, CD133-dependent mechanisms have been shown to be involved in the development of melanoma resistance to chemotherapy [9].

The contribution of CD133 to the regulation of CSCs functions such as self-renewal, differentiation, and drug resistance are likely mediated by the NRTK, Fyn-dependent mechanism via the phosphorylation of Tyr828 residue located on the cytoplasmic domain of CD133 [124-126]. Our laboratory has demonstrated that the phosphorylation of Tyr828 is essential to trigger the activation of PI3K and its downstream dependent signaling pathways in melanoma [9].

The PI3K/AKT pathway is one of the most important networks with the highest mutation frequency in human cancers [125]. Both PI3K/AKT/NF-κB and PI3K/AKT/ mTOR are the two main mutated pathways involved in apoptosis and tumorigenesis facilitating the development of tumor resistance to anti-cancer agents [127]. Dysregulation of major key molecules of these signaling pathways is associated with drug resistance and melanoma progression. Also, elevated activation of PI3K pathway has been suggested to trigger melanoma progression by the activation of PI3K/AKT/ NF-κB axis. [128-130].

In addition to the key role of CD133 in chemoresistance, we and others have demonstrated the cellular mechanisms by which the CD133 protein triggers activation of PI3K pathway both in melanoma and glioblastoma [9, 131].

The PI3K/AKT/mTOR signal to downstream proteins leads to the development of tumor resistance [132, 133]. Also, the PI3K/AKT/mTOR pathway has been shown to play a crucial role in a variety of biological and physiological processes including survival, growth, transcription, and translation, which are associated with the development of drug resistance [134, 135]. Abnormal activation of the PI3K/AKT/mTOR pathway in different tumor types including melanoma has been suggested to be the key mechanism through which tumors evade drug toxicity [136, 137]. Thus, CD133 -mediated activation of PI3K/AKT pathway in melanoma [9] and glioma [138] cells and contributes to the activation of mTORC1 through the phosphorylation of mTOR at Ser248.

In addition to the frequent mutation to AKT family members, mutations to the PI3K-AKT-mTOR pathway are common in melanoma [139, 140]. Mutation in PTEN has been shown to effectively restrain PI3K/AKT/mTOR growth-promoting signaling cascade in primary and metastatic melanoma patients [141, 142]. Upon activation of PI3K or AKT the expression of mTORC1 increases, while its subsequent activation of mTORC1 results in the phosphorylation of the downstream molecules, p70S6K1, and eukaryotic initiation factor 4EBP1, which, in turn, affect mRNA translation and protein synthesis [140-143]. Thus, the key mechanism for the regulation of both PI3K/AKT and PI3K/NF-κB pathways is mediated by Fyn-stimulated CD133 signaling to PI3K [125, 144-146]. To that end, Fyn-stimulated CD133 signaling to PI3K is responsible for the activation of the mTOR allowing melanoma progression [147, 148] drug resistance [149] and recurrence [150]. The mechanisms through which Fyn-stimulated CD133 signal leading to the activation of to PI3K and its downstream pathways are outlined in detail (Fig. 5).] has been added to the main text of the manuscript.

Comment: How about clinical trials of CD133 in melanoma or other cancers? 

What did you learn from this review? What are key questions do you think worth studying? 

Please write the prospects and challenges of CD133.

Authors’ response: Thank you very much for comment. The CD133 protein is a signal mediator. However, the inhibition of tyrosine kinases such as Non-Receptor Tyrosine kinase, Fyn may be of great interest for the inhibition of CSCs growth and resistance. Since the function of CD133 protein as a signal mediator can be enhanced through the phosphorylation of tyrosine 828 and 852 residues located on its cytoplasmic domain. Therefore, the inhibition of Fyn specifically by small molecular inhibitor in the form of kinase inhibitor may have therapeutic impact on the treatment of different tumor types of particularly melanoma.

Comment: The conclusion should be elaborated, better mentioning something past present, and future of CD133.

Authors’ response: Thank you very much for your valuable comment. Accordingly, we improved the conclusion.

See lines: 523-536; the following paragraph [Malignant melanoma is a deadly disease with a poor prognosis. Obtaining a complete tumor remission is difficult because of the presence of heterogeneous subpopulation of CSCs. However, the identification of CSCs in melanoma and other cancers has led to promising advances that may soon impact the management of these cancers. CSCs are also responsible for therapeutic resistance that leads to tumor relapse. Specific signaling mechanisms are required for the maintenance of CSCs in tumors that can maintain their microenvironment. CSCs are becoming priority targets for the development of novel antitumor therapy. The tumor milieu is a critical regulator of melanoma specific CSCs-driven angiogenesis and metastasis. Signaling effectors from ECM or stromal cells can act as EMT or MET inducer or may regulate dormancy at metastatic sites in CSCs. CSC-dependent melanoma progression is mediated by MAPK/ERK, PI3K/Akt/mTOR pathways. Given the unique biology of CSCs, there is great need to develop novel and promising approaches for CSCs targeted cancer therapy. Combinatorial and/or sequential inhibition of CD133 signaling to PI3K/AKT/mTOR and PI3K/RAS/RAF/MEK/ERK pathways, after first-line immunotherapy, may extend the anti-tumor response for melanoma patients. This is especially true for those harboring genetic alterations in key molecules of both PI3K/AKT/mTOR and MAPK/MEK/ERK pathways. The development of clinically relevant pharmacological inhibitors to block the function of both PI3K/AKT/mTOR and MAPK/MEK/ERK could provide new avenues for well-designed studies that assess the tolerability and efficacy of a new therapeutic approach for the treatment of melanoma.]

Comment:The Quality of English Language NA

Authors’ response: Thank you very much for your valuable comment. Accordingly, we improved the quality of English language.

Reviewer 3 Report

Comments and Suggestions for Authors

Review of

CD133-dependent Activation of PI3K PI3K/AKT/mTOR Signaling in Melanoma Progression and Treatment Resistance. 

Overall comments:

This review does not specifically review melanoma instead it generally reviews cancer stem cells, CD133’s role in cancer, CD133, NRTKs, and various signal transduction pathways.

In conclusion, this review is very disjointed on lacks overall structure focusing on the role of these factors in “Melanoma Progression and Treatment Resistance” as stated in the title.

Many references do not mention melanoma but other cancer types although in the text there is a link to melanoma.

Please correct / clarify the points below:

Title:

Is there a repeat of “PI3K” ?

It should be: CD133-dependent Activation of PI3K/AKT/mTOR Signaling in Melanoma Progression and Treatment Resistance.

Figure 1:

Please correct malignat -> malignant

Figure 2

This figure describes the general tumor plasticity, nothing about melanoma.

Differentiation and de-differentiation are switched.

Melanocytes are not epithelial cells, So EMT should not play a role here. Please remove.

Chapter 3, Figure 3:

The title is “Melanoma Stem cells” but it is just a general description of cancer stem cells. The melanocyte lineage should be reviewed in this chapter and figure. 

Chapter 6

The authors review all NRTKs in general but do not focus on the ones relevant to the activation of PI3K in melenoma.

Line 312:

Since PI3K in melanoma is the topic of this review, which NRTK activates PI3K?

Figure 6 legend, Line 317,

It is Abl kinase with an “L”

Line 424:

Figure 8 is not melanoma-specific but it is the general signal transduction pathways downstream of RTKs.

Line 440:

References 164 (about CRC), and 165 (about breast cancer) do not refer to melanoma plaques in the epidermis

Figure 9:

Many of these drugs are not used in melanoma treatment but are inhibitors used in general cancer treatments.

Author Response

Editor

Cells

Dear Editor,

Thank you very much for the encouraging comment regarding our Manuscript ID: ijms-2676058; “CD133-dependent Activation of PI3K PI3K/AKT/mTOR Signaling in Melanoma Progression and Drug resistance.”

As required, enclosed find please our response “Point-for-Point” to the valuable comments of

Reviewer 3

Comments and Suggestions for Authors

Review of CD133-dependent Activation of PI3K PI3K/AKT/mTOR Signaling in Melanoma Progression and Treatment Resistance. 

Overall comments:

This review does not specifically review melanoma instead it generally reviews cancer stem cells, CD133’s role in cancer, CD133, NRTKs, and various signal transduction pathways.

In conclusion, this review is very disjointed on lacks overall structure focusing on the role of these factors in “Melanoma Progression and Treatment Resistance” as stated in the title.

Authors’ response: Thank you very much for your comment. We modified the text to be more specific.

Comment: Many references do not mention melanoma but other cancer types although in the text there is a link to melanoma.

Authors’ response: Thank you very much for your valuable comment. Accordingly, we adjusted the references over all in the manuscript.

Please correct / clarify the points below:

Comment: Title: Is there a repeat of “PI3K”? It should be: CD133-dependent Activation of PI3K/AKT/mTOR Signaling in Melanoma Progression and Treatment Resistance.

Authors’ response: Thank you very much for your comment. We improved the manuscript to meet your request.

Comment: Figure 1: Please correct malignat -> malignant

Authors’ response: Thank you very much for your comment. As required, we corrected figure 1.

See Figure 1

Comment: Figure 2. This figure describes the general tumor plasticity, nothing about melanoma. Differentiation and de-differentiation are switched. Melanocytes are not epithelial cells, So EMT should not play a role here. Please remove.

Authors’ response: Thank you very much for comment. Accordingly, we modified Fig. 2 and the corresponding legend.

See Lines:183-187; the following ligand to figure 2 [Figure 2. Mechanisms of melanoma plasticity. Melanoma plasticity is a process whereby melanoma cells with intraturmoral heterogeneity shift dynamically and reversibly between cancer stem cells (CSCs)/intermediate state and non-cancer stem cells (nCSCs)/differentiated state. Melanoma cell plasticity is mediated by an epithelial-to-mesenchymal transition (EMT) program of CSCs in which CSCs serve as an intermediate state. The mechanisms of tumor cell plasticity are associated with genetic mutations, epigenetic modifications, and transcriptional modulation of cancer cells and TME-dependent signals (i.e., growth factors, cytokines, CAFs or TAMs) in response to inflammation, injury, and senescence.] by the following ligand [Figure 2. Mechanisms of melanoma plasticity. Melanoma cell plasticity is the ability of tumor cells to shift dynamically between differentiated/cancer stem cells (CSCs) and undifferentiated /non-cancer stem cells (non-CSCs) states to promote long-term tumor cell growth, invasiveness, and migration. The mechanisms of melanoma plasticity are mediated by genetic and epigenetic modifications of melanoma cells and by significant alteration of tumor microenvironmental secretory products (e.g., growth factors and cytokines), activation of cancer-associated fibroblasts (CAFs) or tumor-associated macrophages (TAMs).].

Comment: Chapter 3, Figure 3: The title is “Melanoma Stem cells” but it is just a general description of cancer stem cells. The melanocyte lineage should be reviewed in this chapter and figure. 

Authors’ response: Thank you very much for your valuable commen

See Lines:190-203; The following paragraph [The processes of melanoma development start in mature melanocytes [91]. Although accumulated evidence indicates that early-stage precursors of melanocytes exist in the dermis [92-94], while other reports suggested that earliest origins of cutaneous melanoma may have occurred in extrafollicular melanocyte stem cells [95-97]. It is probably that melanoma stem cells (MSCs) are derived either from a transformed melanocyte, from a transformed melanocyte stem cell, or from a combination of both sources. Like CSCs, MSCs are characterized by their ability to self-renew and differentiate [4, 5, 9]. The generation of these subpopulations is mediated via genetic segregation and epigenetic alterations via transcriptional regulation of genes associated with stemness properties and dysregulation of aberrant signaling pathways such as FOXM1 signaling [98]. Consequently, melanoma cells undergo intrinsically asymmetric cell divisions of stem cell lineage to produce two daughter cells, both significantly different in their genetic material and phenotype [99, 100]. One of these daughter cells is characterized by its stemness properties and is referred to as CSC, whereas the other daughter cell (non-CSC) represents a larger portion of the overall tumor mass [101]. In addition to the deregulation of aberrant signaling pathways for tumor growth and survival, MSCs are notable for their expression of CD20, CD133, CD166, CD271, ABCB5, Nestin and CD105 surface markers [4,5, 9]. Thus, understanding the pathways controlling self-renewal, expansion, and differentiation of MSCs and how UV radiation alters and disrupts melanocyte lineage pathways will bring greater insight into the origins of melanomas.] has modified.

References

See Lines: 781-795; the following references [Hoerter, J. D.; Bradley, P.; Casillas, A.; Chambers, D.; Weiswasser, B.; Clements, L.; Gilbert, S.; Jiao, A. Does melanoma begin in a melanocyte stem cell? J Skin Cancer 20122012, 571087. DOI: 10.1155/2012/571087.

Grichnik, J. M.; Ali, W. N.; Burch, J. A.; Byers, J. D.; Garcia, C. A.; Clark, R. E.; Shea, C. R. KIT expression reveals a population of precursor melanocytes in human skin. J Invest Dermatol 1996106 (5), 967-971. DOI: 10.1111/1523-1747.ep12338471.

Li, L.; Fukunaga-Kalabis, M.; Herlyn, M. Isolation and cultivation of dermal stem cells that differentiate into functional epidermal melanocytes. Methods Mol Biol 2012806, 15-29. DOI: 10.1007/978-1-61779-367-7_2.

Zabierowski, S. E.; Fukunaga-Kalabis, M.; Li, L.; Herlyn, M. Dermis-derived stem cells: a source of epidermal melanocytes and melanoma? Pigment Cell Melanoma Res 201124 (3), 422-429. DOI: 10.1111/j.1755-148X.2011.00847.x.

Nanda, V. G. Y.; Peng, W.; Hwu, P.; Davies, M. A.; Ciliberto, G.; Fattore, L.; Malpicci, D.; Aurisicchio, L.; Ascierto, P. A.; Croce, C. M.; et al. Melanoma and immunotherapy bridge 2015 : Naples, Italy. 1-5 December 2015. J Transl Med 201614 (1), 65. DOI: 10.1186/s12967-016-0791-2.

Kyrgidis, A.; Tzellos, T. G.; Triaridis, S. Melanoma: Stem cells, sun exposure and hallmarks for carcinogenesis, molecular concepts and future clinical implications. J Carcinog 20109, 3. DOI: 10.4103/1477-3163.62141.

Valyi-Nagy, K.; Kormos, B.; Ali, M.; Shukla, D.; Valyi-Nagy, T. Stem cell marker CD271 is expressed by vasculogenic mimicry-forming uveal melanoma cells in three-dimensional cultures. Mol Vis 201218, 588-592.] have been added to the references section.

Comment: Chapter 6: The authors review all NRTKs in general but do not focus on the ones relevant to the activation of PI3K in melanoma.

Authors response: Thank you very much for your comment. As required, we added a paragraph to the text of the manuscript describing the structure and the regulation of NRTK, Fyn. Fyn is responsible for the phosphorylation of tyrosine kinase residues located on the cytoplasmic domain of CD133 to mediate the activation of PI3K.

See lines: 343-361; the following text [7. Non-receptor tyrosine kinase Fyn

Fyn is a tyrosine specific phospho-transferase that is a member of the large Src family of NRTKs. While the formal crystal structure of the full-length Fyn protein has not been described, the mode of regulation of Fyn tyrosine kinase activity is like the Src family kinases [187]. Fyn is 59kDa protein comprised of 537 amino acids encoded by the Fyn gene that can be spliced to produce three isoforms [188]. The first identified isoform is the isoform 1 (Fyn[B]), isoform 2 (Fyn [T]) is highly expressed in T-cells and differs from isoform in the linker region between the SH2 and SH1 domains [189]. Isoform 3 has been detected in blood cells and differs from isoform 1 via the absence of sequence 233-287 [190]. Like members of Src family, Fyn shares the conserved structure that consists of consecutive SH1, SH2 and SH3 domains (Fig. 7). The SH1 domain is the catalytic tyrosine kinase, while the SH2 domain binds to tyrosine-phosphorylated substrates. Specifically, the SH2 domain of Fyn binds the phosphorylated tyrosine Y528 residue in its carboxyl terminal tail under basal conditions in vivo [191, 192]. Repression of Fyn kinase activity is achieved via intra-molecular interactions between the SH3 domain and a polyproline type II linker helix that connects the SH2 and the SH1 domains. For Fyn kinase, the tyrosine Y528 negative regulatory site is phosphorylated by C-terminal src Kinase (Csk), a cytoplasmic protein-tyrosine kinase [193]. Csk homology kinase (CHK) is a second enzyme that catalyzes the phosphorylation of this inhibitory tyrosine Y528 [194]. While Csk is expressed in all mammalian cells, CHK expression is limited to breast, haematopoietic cells, neurons, and testes. CHK binds Src family members with a high affinity, independent of CHK catalytic activity, which may be sufficient to inhibit Src family kinase activity [194]. The dephosphorylation of the Y528 residue by protein tyrosine phosphatases rPTPα, SHP1/2, PTP1B, PTPε and CD45 can release the SH2 domain and activate the enzyme [195-199]. In addition, the subfamily composed of Fyn, Src and Lyn kinases contains dual acylation sites in the amino-terminal SH4 domain, which is thought to be partially responsible for lipid raft micro-domain association [200].]. To the main text of the manuscript.

Figures

A new Figure has been created; See Fig. 7.

Legends to Figures

See lines: 363-365; the following legend [Fig. 7. Fyn kinase structure and regulation. Fyn kinase consists of SH1, SH2, SH3 and SH4 domains. The SH2 domain binds the phosphorylated Tyr528 (pTyr528) in the C-terminus to keep Fyn in an inactive conformation. Tyr528 is dephosphorylated by phosphatases (PTPs) to keep the structure open allowing for phosphorylation of Tyr416 in the catalytic SH1 domain. ] has been added to the figure. 7.

References

See Lines:1019-1050; the following references [Parsons, S. J.; Parsons, J. T. Src family kinases, key regulators of signal transduction. Oncogene 200423 (48), 7906-7909. DOI: 10.1038/sj.onc.1208160.

Cooke, M. P.; Perlmutter, R. M. Expression of a novel form of the fyn proto-oncogene in hematopoietic cells. New Biol 19891 (1), 66-74.

Thomas, S. M.; Brugge, J. S. Cellular functions regulated by Src family kinases. Annu Rev Cell Dev Biol 199713, 513-609. DOI: 10.1146/annurev.cellbio.13.1.513.

Goldsmith, J. F.; Hall, C. G.; Atkinson, T. P. Identification of an alternatively spliced isoform of the fyn tyrosine kinase. Biochem Biophys Res Commun 2002298 (4), 501-504. DOI: 10.1016/s0006-291x(02)02510-x.

Zheng, X. M.; Resnick, R. J.; Shalloway, D. A phosphotyrosine displacement mechanism for activation of Src by PTPalpha. EMBO J 200019 (5), 964-978. DOI: 10.1093/emboj/19.5.964.

Sicheri, F.; Kuriyan, J. Structures of Src-family tyrosine kinases. Curr Opin Struct Biol 19977 (6), 777-785. DOI: 10.1016/s0959-440x(97)80146-7.

Nada, S.; Okada, M.; MacAuley, A.; Cooper, J. A.; Nakagawa, H. Cloning of a complementary DNA for a protein-tyrosine kinase that specifically phosphorylates a negative regulatory site of p60c-src. Nature 1991351 (6321), 69-72. DOI: 10.1038/351069a0.

Chong, Y. P.; Mulhern, T. D.; Zhu, H. J.; Fujita, D. J.; Bjorge, J. D.; Tantiongco, J. P.; Sotirellis, N.; Lio, D. S.; Scholz, G.; Cheng, H. C. A novel non-catalytic mechanism employed by the C-terminal Src-homologous kinase to inhibit Src-family kinase activity. J Biol Chem 2004279 (20), 20752-20766. DOI: 10.1074/jbc.M309865200.

Chan, A. C.; Desai, D. M.; Weiss, A. The role of protein tyrosine kinases and protein tyrosine phosphatases in T cell antigen receptor signal transduction. Annu Rev Immunol 199412, 555-592. DOI: 10.1146/annurev.iy.12.040194.003011.

Asante-Appiah, E.; Kennedy, B. P. Protein tyrosine phosphatases: the quest for negative regulators of insulin action. Am J Physiol Endocrinol Metab 2003284 (4), E663-670. DOI: 10.1152/ajpendo.00462.2002.

Chong, Y. P.; Ia, K. K.; Mulhern, T. D.; Cheng, H. C. Endogenous and synthetic inhibitors of the Src-family protein tyrosine kinases. Biochim Biophys Acta 20051754 (1-2), 210-220. DOI: 10.1016/j.bbapap.2005.07.027.

Poole, A. W.; Jones, M. L. A SHPing tale: perspectives on the regulation of SHP-1 and SHP-2 tyrosine phosphatases by the C-terminal tail. Cell Signal 200517 (11), 1323-1332. DOI: 10.1016/j.cellsig.2005.05.016.

Roskoski, R. Src kinase regulation by phosphorylation and dephosphorylation. Biochem Biophys Res Commun 2005331 (1), 1-14. DOI: 10.1016/j.bbrc.2005.03.012.

Boggon, T. J.; Eck, M. J. Structure and regulation of Src family kinases. Oncogene 200423 (48), 7918-7927. DOI: 10.1038/sj.onc.1208081.

Rezatabar, S.; Karimian, A.; Rameshknia, V.; Parsian, H.; Majidinia, M.; Kopi, T. A.; Bishayee, A.; Sadeghinia, A.; Yousefi, M.; Monirialamdari, M.; et al. RAS/MAPK signaling functions in oxidative stress, DNA damage response and cancer progression. J Cell Physiol 2019234 (9), 14951-14965. DOI: 10.1002/jcp.28334.] have been added to the references section.

Comment: Line 312: Since PI3K in melanoma is the topic of this review, which NRTK activates PI3K?

Authors’response: Thank you very much for your comment. Fyn is the non -receptor tyrosine kinase, which is responsible for the activation of PI3K via CD133-dependent mechanism. We added a paragraph describing the structure and regulation of Fyn. See lines: 363-365 and Fig. 7

Comment: Figure 6 legend, Line 317,

It is Abl kinase with an “L”

Authors’ response: Thank you very much for your comment. Accordingly, we corrected Abl over on the text of the manuscript.

See lines: 310, 311 and 321

Comment: Line 424: Figure 8 is not melanoma-specific, but it is the general signal transduction pathways downstream of RTKs.

Authors’ response: Thank you very much for your valuable comment. First Fig. 8 becomes Fig. 9. Accordingly, we modified the legend of figure 9. as required,

See lines: 451-456; the following legend [ Figure 9. Downstream effectors of RTKs -mediated pathways and their cellular functions. The activation of PI3K/AKT and NRAS/RAF/MEK signaling pathways by RTKs can promote several cellular functions via the activation of MDM2, NF-κB, and CREB to enhance p53 degradation as well as promote the transcription of proliferative and anti-apoptotic genes. PI3K/AKT-induced activation of mTOR enhances the activation of both 4EBP1 and p70SK to induce the translation of angiogenic or cell cycle gene targets. PI3K/AKT can inhibit the phosphorylation of downstream effectors, namely caspase-9, GSK-3ß, p27, and p21 to induce cell cycle progression and inhibition of apoptosis.] has been modified.

Comment: Line 440: References 164 (about CRC), and 165 (about breast cancer) do not refer to melanoma plaques in the epidermis

Authors’ response: Thank you very much for your comment. As required, we replaced the mentioned references with new references citing Melanoma.

Comment: Figure 9: Many of these drugs are not used in melanoma treatment but are inhibitors used in general cancer treatments.

Authors’ response: Thank you very much for your comment. Accordingly, we proved the mentioned inhibitors and cited the already used for melanoma.
